# Expression of the Reverse Transcriptase Domain of Telomerase Reverse Transcriptase Induces Lytic Cellular Response in DNA-Immunized Mice and Limits Tumorigenic and Metastatic Potential of Murine Adenocarcinoma 4T1 Cells

**DOI:** 10.3390/vaccines8020318

**Published:** 2020-06-18

**Authors:** Juris Jansons, Ekaterina Bayurova, Dace Skrastina, Alisa Kurlanda, Ilze Fridrihsone, Dmitry Kostyushev, Anastasia Kostyusheva, Alexander Artyuhov, Erdem Dashinimaev, Darya Avdoshina, Alla Kondrashova, Vladimir Valuev-Elliston, Oleg Latyshev, Olesja Eliseeva, Stefan Petkov, Maxim Abakumov, Laura Hippe, Irina Kholodnyuk, Elizaveta Starodubova, Tatiana Gorodnicheva, Alexander Ivanov, Ilya Gordeychuk, Maria Isaguliants

**Affiliations:** 1Department of Research, and Department of Pathology, Pathology, Rīga Stradiņš University, LV-1007 Riga, Latvia; juris.jansons@rsu.lv (J.J.); alisa.kurlanda@rsu.lv (A.K.); ilze.fridrihsone@rsu.lv (I.F.); laura.hippe@rsu.lv (L.H.); irina.holodnuka@rsu.lv (I.K.); 2Latvian Biomedical Research and Study Centre, LV-1067 Riga, Latvia; daceskr@biomed.lu.lv; 3N.F. Gamaleya National Research Center for Epidemiology and Microbiology, Moscow 127994, Russia; 79153645941@ya.ru (E.B.); oleglat80@mail.ru (O.L.); olesenka80@mail.ru (O.E.); abakumov1988@gmail.com (M.A.); aivanov@yandex.ru (A.I.); lab.gord@gmail.com (I.G.); 4Chumakov Federal Scientific Center for Research and Development of Immune-and-Biological Products of Russian Academy of Sciences, Moscow 127994, Russia; darya_avdoshina@mail.ru (D.A.); varyaw96@gmail.com (A.K.); 5National Medical Research Center of Tuberculosis and Infectious Diseases, Ministry of Health, Moscow 127994, Russia; dkostushev@gmail.com (D.K.); kostyusheva_ap@mail.ru (A.K.); 6Center for Precision Genome Editing and Genetic Technologies, Pirogov Russian National Research Medical University, Moscow 127994, Russia; alexanderartyuhov@gmail.com (A.A.); dashinimaev@gmail.com (E.D.); 7Koltzov Institute of Developmental Biology of Russian Academy of Sciences, Moscow 127994, Russia; 8Engelhardt Institute of Molecular Biology, Russian Academy of Sciences, Moscow 127994, Russia; gansfaust@mail.ru (V.V.-E.); estarodubova@yandex.ru (E.S.); 9Department of Microbiology, Tumor and Cell Biology, Karolinska Institutet, 17177 Stockholm, Sweden; Stefan.petkov@ki.se; 10Laboratory of Biomedical Nanomaterials, National University of Science and Technology MISIS, Moscow 127994, Russia; 11Department of Medical Nanobiotechnologies, Pirogov Russian National Research Medical University, Moscow 127994, Russia; 12Evrogen, Moscow 127994, Russia; tatiana.gorod@evrogen.ru; 13Institute for Translational Medicine and Biotechnology, Sechenov First Moscow State Medical University, Moscow 127994, Russia

**Keywords:** therapeutic cancer vaccines, telomerase reverse transcriptase (TERT), reverse transcriptase domain, intradermal DNA immunization, electroporation, epitopes, CD4+ and CD8+ lytic T cell response, antibodies, murine adenocarcinoma cells, lentiviral transduction, tumor growth, suppression, rejection, metastasis

## Abstract

Telomerase reverse transcriptase (TERT) is a classic tumor-associated antigen overexpressed in majority of tumors. Several TERT-based cancer vaccines are currently in clinical trials, but immune correlates of their antitumor activity remain largely unknown. Here, we characterized fine specificity and lytic potential of immune response against rat TERT in mice. BALB/c mice were primed with plasmids encoding expression-optimized hemagglutinin-tagged or nontagged TERT or empty vector and boosted with same DNA mixed with plasmid encoding firefly luciferase (Luc DNA). Injections were followed by electroporation. Photon emission from booster sites was assessed by in vivo bioluminescent imaging. Two weeks post boost, mice were sacrificed and assessed for IFN-γ, interleukin-2 (IL-2), and tumor necrosis factor alpha (TNF-α) production by T-cells upon their stimulation with TERT peptides and for anti-TERT antibodies. All TERT DNA-immunized mice developed cellular and antibody response against epitopes at the N-terminus and reverse transcriptase domain (rtTERT) of TERT. Photon emission from mice boosted with TERT/TERT-HA+Luc DNA was 100 times lower than from vector+Luc DNA-boosted controls. Bioluminescence loss correlated with percent of IFN-γ/IL-2/TNF-α producing CD8+ and CD4+ T-cells specific to rtTERT, indicating immune clearance of TERT/Luc-coexpressing cells. We made murine adenocarcinoma 4T1luc2 cells to express rtTERT by lentiviral transduction. Expression of rtTERT significantly reduced the capacity of 4T1luc2 to form tumors and metastasize in mice, while not affecting in vitro growth. Mice which rejected the tumors developed T-cell response against rtTERT and low/no response to the autoepitope of TERT. This advances rtTERT as key component of TERT-based therapeutic vaccines against cancer.

## 1. Introduction

Cancer immunotherapies fall in two major categories: passive that includes blockade of immune checkpoints and adoptive immunotherapy and active that includes therapeutic vaccination and immuno/chemotherapy combinations [1]. Multiple tumor-associated antigens (TAAs) expressed by cancer cells have been identified as targets of functional anticancer T-cell response and tested as immunogens for active cancer immunotherapy [2,3]. Active immunotherapies based on TAAs presented by the patient autologous tumor cells or dendritic cells (DCs) as well as chimeric antigen receptor T cell immunotherapy (CAR-T) are effective, specifically if combined with surgery and radio- and/or chemotherapy [4,5,6]. However, design and preparation of such vaccines as well as treatments are individualized, which implies extra time and extra costs. Straightforward active tumor immunotherapy implying immunization with TAA-based vaccines is still underdeveloped. Classic vaccination with recombinant antigens or peptides as monotherapy gives little clinical benefit [1,7]. Genetic vaccines show better clinical efficacy [8,9], also in veterinary [10,11] and translational oncology [12]. Potency of DNA immunization was demonstrated in a variety of clinical applications [13,14]. The first successful therapeutic DNA vaccine reported to cause recession of cervical neoplasia based on the consensus tumor antigens of HPV16 [15] is currently in two Phase III clinical trials (NCT03185013 and NCT03721978). Development of other candidate cancer DNA vaccines based on endogenous and viral antigens is in progress [16].

Telomerase reverse transcriptase (TERT) is a classic TAA overexpressed in majority of tumors [17]. The telomerase enzyme catalyzes de novo synthesis of telomere repeats, maintaining telomere length required for unlimited cell proliferation. Human telomere synthesis occurs early in development [18,19]. The majority of adult somatic cells do not have appreciable telomerase activity and telomeres gradually shorten, limiting cell division capacity [20]. In the majority of human cancers, however, telomerase is reactivated and supports the sustained proliferative capacity of these cells [21]. This reactivation could be induced by DNA damage [22] and/or deregulated expression of the MYC oncoprotein [23]. Catalytic component of telomerase—telomerase reverse transcriptase (TERT) is the rate-limiting factor for telomerase activity; it is expressed in virtually all tumors [17]. Degraded fragments of TERT expressed on the surface of tumor cells are recognized by the immune system and induce cellular immune response resulting in a partial control of tumor growth [24]. Overexpressed in tumors and immunogenic in subjects controlling tumor growth, TERT presents as an attractive target of therapeutic cancer vaccines. In addition, TERT is highly conserved, also in tumors, with cancer-associated mutations occurring mainly in the promoter region [25], which opens a possibility to make this vaccine basically universal. In vivo studies have revealed both safety and curative potential of TERT-based immunotherapy, which does not harm healthy cells [26,27]. First clinical trials of DNA vaccine based on TERT applied in cancer patients reported it to be safe, well tolerated, and immunogenic [28,29], immunization resulting in disease stabilization in 58% of patients with relapsed or refractory cancers [28]. Despite this promising progress, the panel of TERT-based vaccine candidates is still limited, and clinical trials are few (NCT04280848; NCT00510133; NCT00753415, NCT00961844, NCT01153113, NCT01935154, NCT02293707, NCT02960594, NCT03265717, and NCT03502785; and NCT03491683 and NCT03946358). This motivates intensive efforts to create a wider panel of efficacious TERT-based tumor vaccines as well as delineate the immune mechanism(s) underlying their efficacy.

Here, we have designed a new variant of cancer vaccine against TERT, an expression-optimized DNA encoding rat TERT. Our experiments demonstrated that intradermal injection of TERT DNA into mice followed by electroporation induces an effector response of CD8+ and CD4+ T cells against multiple epitopes, specifically in the reverse transcriptase (RT) domain of TERT (rtTERT). Expression of this domain by murine adenocarcinoma cells drastically reduced their capacity to form tumors and generate metastasis. Mice with restricted growth of rtTERT-expressing tumor cells exhibited CD4+ and CD8+ T cell response against epitopes of rtTERT that are recognized by DNA-immunized mice. We also mapped autoepitopes of TERT lying outside rtTERT and demonstrated that T-helper cell response against such epitopes promotes tumor growth. This points at RT domain as the necessary and sufficient as well as safe component of therapeutic cancer vaccines based on TERT.

## 2. Materials and Methods

### 2.1. Plasmids

The amino acid sequence of rat TERT was taken from UniProtKB database (accession number Q673L6). Two variants of rat TERT gene were used in the study, one—encoding the native protein (TERT) and second—recombinant construct with hemagglutinin (HA) tag at the C-terminus (TERT-HA). TERT and TERT-HA proteins were encoded by synthetic genes optimized for expression in mammalian cells (GenBank submissions MK749423 and MK749424). To increase protein expression, coding sequences were supplied with the Kozak sequence AAT-ATG-GGA at the 5′-end resulting in Met-Gly at the protein N-terminus. TERT-encoding sequences were cloned into the eukaryotic expression vector pVAX1 (Invitrogen, Waltham, MA, USA) generating two plasmids—pVax-TERT and pVax-TERT-HA. Nucleotide sequence encoding RT domain of rat TERT (rtTERT) was amplified using primers 5′-AGGAATAACATATGGGAGTGAA-3′ and 5′-AATGGATCCTTACAAGCCACACCAGGGAAAC-3′ and cloned into prokaryotic expression vector pET15b using NdeI and BamHI sites, generating plasmid pET15rtTERT. To follow gene expression after DNA immunization, we used the plasmid encoding firefly luciferase pVaxLuc2 (kind gift of A.K. Roos, Karolinska Institutet, Stockholm, Sweden). Plasmids were produced in *E. coli* and purified using Plasmid EndoFree Kits (Qiagen, Hilden, Germany) as recommended by the manufacturer.

### 2.2. Peptides and Recombinant Proteins Used for Immunoassays

TERT-derived peptides used in the assays of cellular and antibody immunogenicity are listed in Table 1. Peptides (SynPep Ltd., Shanghai, China) were purified by HPLC to 70% purity; their structure was confirmed by mass spectrometry.

Anti-TERT antibody ELISA was performed using recombinant His-tagged rtTERT produced in *E. coli*. For this, *E. coli* Rosetta (DE3) strain (Novagen, Darmstadt, Germany) harboring extra copies of tRNAs, rarely used in *E coli*, were transformed with pET15rtTERT. Recombinant rtTERT was expressed as described previously for hepatitis C virus NS5A [40]. Protein purification was performed in the denaturing conditions by affinity chromatography on Ni-NTA-agarose (Novagen); buffers were supplemented with Protease Inhibitor Cocktail for use in purification of His-tagged proteins as recommended by the manufacturer (Sigma, Darmstadt, Germany). Fraction containing the protein with molecular mass corresponding to rtTERT was identified after analysis of fractions on 12% SDS-PAGE followed by Coomassie R250 staining. The protein was renatured by dialysis in three steps: initially, against buffer A (20 mM Tris-HCl, pH 7.5, 500 mM NaCl, 4 M urea, and 10% glycerol), then against buffer A (100 mL) to which 500 mL buffer B (20 mM Tris-HCl, pH 7.5, 1 M NaCl and 10% glycerol) was slowly added, and finally, against buffer C (20 mM Tris-HCl, pH 7.5, 500 mM NaCl, and 50% glycerol). Purified rtTERT was aliquoted and stored at −20 °C.

### 2.3. Transient Expression of TERT in Mammalian Cells

Briefly, 293T cells were cultivated in high-glucose DMEM (PanEco, Moscow, Russia) supplemented with 10% fetal bovine serum (FBS)(HyClone, GE Healthcare, Chicago, IL, USA) and penicillin/streptomycin (both from HyClone). Cells were transfected with plasmids pVAX1, pVax-TERT, or pVax-TERT-HA using Lipofectamine LTX (Invitrogen, Waltham, MA, USA) according to the manufacturer’s instructions. Two days later, cells were harvested, lysed, and subjected to electrophoreses in 10% PAAG followed by Western blot using protocols described previously [41]. Blots were stained with polyclonal rabbit antibodies against synthetic peptide derived from TERT (Ab191523, Abcam, Cambridge, UK) diluted 1:1000, stripped, and then, re-stained with monoclonal anti-actin antibodies (Sigma-Aldrich, St. Louis, MO, USA) diluted 1:5000.

### 2.4. Generation of 4T1luc2 Derivatives Expressing rtTERT by Lentiviral Transduction

Derivatives of murine mammary gland adenocarcinoma cells expressing firefly luciferase 4T1luc2 (“Bioware Ultra Cell Line 4T1luc2,” Caliper, Hopkinton, MA, USA; http://www.caliperls.com/assets/014/7158.pdf) were generated as described previously [42]. In brief, coding sequence for rtTERT was recloned from pET15rtTERT into lentiviral vector pRRLSIN.cPPT.PGK (Addgene plasmid #12252; a gift from D. Trono) under the control of the human phosphoglycerate kinase (PGK) promoter, generating lentiviral vector pLVrtTERT (Appendix A). Lentiviral particles were produced by transient transfection of 293T cells as described elsewhere [43] and concentrated 10-fold with Amicon Ultra-15 100 K centrifuge concentrators (Merck-Millipore, Darmstadt, Germany). Infectious titers of the lentiviral particles were determined in HT1080 cells by quantitative real-time PCR [43] using standard samples of HT-1080 DNA with a known number of viral genome copies. Transduction of 4T1luc2 cells was performed with the multiplicity of lentiviral infection of 5 and 180 transducing units per cell. Monoclonal populations of 4T1luc2 derivative clones were generated by limiting dilution in 96-well plates. Resulting 4T1luc2 derivatives were cultured in the full RPMI-1640 medium with 10% FBS and 100 mg/mL penicillin/streptomycin mix at 37 °C with 5% CO_2_ and split every 2–3 days. Doubling time of derivative clones was estimated as described previously [42]. Presence of inserts of rtTERT DNA was confirmed by PCR and sequencing. Clones were characterized by the number of inserts of rtTERT DNA in the genome of 4Tluc2 cells determined by ddPCR (see below).

### 2.5. Extraction of Nucleic Acids and Analysis of Genomic rtTERT Inserts in 4T1luc2 Daughter Clones

Cell culture medium was discarded and cells were detached using 0.25% trypsin with EDTA before purification with MagNA Pure Compact Nucleic Acid Isolation Kit I—Large Volume (Roche, Basel, Switzerland) using MagNA Pure Compact Instrument (Roche, Basel, Switzerland) according to the manufacturer’s instructions. DNA concentration was determined using Quant-IT™ PicoGreen™ dsDNA Assay Kit (Thermo Fisher, Waltham, MA, USA) on QUANTIFLUOR™-ST fluorometer (Promega, Madison, WI, USA). Presence of inserts encoding rtTERT in the genome of derivative clones was verified by PCR with specific primers (Appendix A). Analysis was performed directly after DNA purification.

Copy number of *rtTERT* loci with respect to invariant reference loci *Mstn* and *Actb* was estimated using digital droplet PCR (ddPCR). Copy number of *rtTERT* inserts was calculated as the number of detected *rtTERT* loci in DNA sample, divided by the number of *Mstn* and *Actb* loci and multiplied by 2 (number of *Mstn* and *Actb* copies). Reaction mixes were prepared using ddPCR EvaGreen Supermix (Bio-Rad, Hercules, CA, USA) using 10 ng of genomic DNA and 250 nM of primers (Appendix A) per reaction. Droplets were generated using automated Droplet Generator (Bio-Rad). Thermocycling was performed on C1000 Touch Thermal Cycler (Bio-Rad), thermal cycling protocol is presented in Appendix A. Data were collected using QX200 Droplet Reader (Bio-Rad) and analyzed using QuantaSoft software version 1.7.4.0917 (Bio-Rad). Results of primer validation are presented in Appendix A. Two clearly distinguishable clusters of positive and negative droplets were observed for *Actb*, *Mstn,* and *rtTERT* (Appendix A, respectively). No significant amplification was observed for any primer pair in the absence of the template (Appendix A).

### 2.6. Reverse Transcription and Analysis of rtTERT mRNA Expression by Semiquantitative PCR

Nucleic acids extracted and purified as described above were reverse transcribed using MMLV reverse transcription kit (Evrogen, Moscow, Russia). Gene-specific PCRs were performed on Applied Biosystems QuantStudio 5 cycler (Thermo Fisher) with SYBR Green Kit (Evrogen) using primers specific to *rtTERT* and presented relative to levels of mRNA of *HPRT1.* Respective primer sequences are presented in Appendix A. Relative gene expression levels were calculated using ddCt method [44].

### 2.7. Analysis of Expression of Endogenous TERT in 4T1luc2 Clones by Immunofluorescent Microscopy

Parental 4T1luc2 cells and daughter clones were assessed for expression of endogenous TERT by immunofluorescence using commercial rabbit anti-TERT antibodies Ab191523 (Abcam). Peptide used to generate Ab191523 localizes outside of rtTERT, hence the antibodies do not recognize the rtTERT domain of rat TERT. Staining was performed as follows. Briefly, 4T1luc2 and derivate clones were seeded on glass coverslips and fixed in 4% paraformaldehyde for 10 min. Next, coverslips were washed 3 times in Tris-HCl (50 mM, pH 7.8), incubated for 30 min with blocking buffer (50 mM Tris-HCl, pH 7.8, 0.02% of Triton X-100, 10% horse sera, and 150 mM NaCl), and incubated with primary antibodies (1:50) for 1 h at 20 °C. Cells were washed 3 times for 5 min in washing buffer (50 mM Tris-HCl, pH 7.8, 0.02% of Triton X-100 and 200 mM NaCl) and then, incubated with secondary Alexa Fluor 488 goat antirabbit IgG antibodies (ab150077, Abcam; 1:350) supplemented with Hoechst 33,342 to visualize the nuclei (1/10,000; Abcam) for 1 h at 20 °C. Coverslips were washed 3 times for 5 min in washing buffer and mounted with Fluoroshield Mounting Medium (Abcam). Images were captured using a Leica DMI6000 microscope with 100× immersion objective and analyzed using ImageJ software (http://rsb.info.nih.gov/ij). Corrected total cell fluorescence (CTCF) was calculated according to M. Fitzpatrick’s protocol (https://theolb.readthedocs.io/en/latest/imaging/measuring-cell-fluorescence-using-imagej.html) as CTCF = Integrated density − (area of selected cell × background fluorescent signal). The research was done using equipment of the Core Centrum of Institute of Developmental Biology (Russian Academy of Sciences, Moscow, Russia).

### 2.8. Assessment of Genetic Stability of 4T1luc2 Clones

To assess genetic stability of rtTERT-expressing 4T1luc2 clones, cells were stained for γ-H2AX foci with anti-γ-H2AX mouse monoclonal antibodies Ab26350 (Abcam) as described previously [45] using secondary Alexa Fluor 594 goat anti-mouse antibodies (ab150116, Abcam) mixed with Hoechst 33,342 for nuclear staining. Images were captured and analyzed as described above for the analysis of expression of endogenous TERT.

### 2.9. Cell Cycle Analysis of 4T1luc2 Clones

At harvest, live cells were washed twice in PBS and incubated with 20 μM DRAQ5 (ab108410, Abcam) for 10 min at 37 °C. Samples were kept in the dark and analyzed on the BD FACSCanto II cytometer (BD Biosciences, Franklin Lakes, NJ, USA). A total of 100,000 cells per sample were used for the analysis. All samples were analyzed in independent triplicates. Live (gate P1) and single (gate P2) cells were gated in NovoExpress Software (ACEA Biosciences, Inc, San Diego, CA, USA) (Appendix A). DRAQ5-stained cells were analyzed at the APC channel. Distribution of cells in SubG1, G1/G0, S, G2/M and SuperG2 areas was assessed using Watson pragmatic algorithm [46] in the NovoExpress software. In control groups, peak G1 was determined manually according to the manufacturer’s recommendations, peak G2 was set at ×1.75 of G1 peak (Appendix A). Further analysis was performed automatically with pre-set G1 and G2 peaks and coefficients of variation (CV) at all samples.

### 2.10. Animals and Animal Experiments

Eight-week-old BALB/c mice from Laboratory Animal Center University of Tartu (Tartu, Estonia) were housed under a 12/12 h light/dark cycle with ad libitum access to water and food. Experiments were carried in compliance with the bioethical principles adopted by the European Convention for the Protection of Vertebrate Animals Used for Experimental and Other Scientific Purposes (Strasbourg, France, 1986). Experimental procedures were approved by the ethics committee of the Latvian Animal Protection Ethics Committee of the Latvian Food and Veterinary Service, permit No 99 from 4 April 2018. Inhalation anesthesia in mice was induced in a ventilated chamber containing air with 4% isoflurane and maintained with 2.5% isoflurane/air mix administered through a facial mask during all intradermal injections and electroporation.

### 2.11. DNA Immunization of Mice

BALB/c mice were primed with two 20 µg doses of either pVax-TERT (*n* = 5), or pVax-TERT-HA (*n* = 5), or empty vector pVAX1 (*n* = 5), and after 3 weeks boosted with 15 µg of the same plasmids mixed with 5 µg of pVaxLuc2 (20 µg of DNA per site in total). At each immunization, mice received two intradermal (id) injections of plasmid DNA solution in PBS delivered to the left and to the right from the back of the tail. Plasmids were administered with 29G-needle insulin syringes. Injections were followed by electroporation using CUY21EditII (BEX Co., Tokyo, Japan) in vivo electroporator with fork-plate (CUY663-5 × 10) electrode (BEX Co., Tokyo, Japan) with a poration pulse of 400 V (0.1 ms with a 20 ms break) followed by 8 altering polarity (+/−) driving pulses of 10 ms performed at 100 V with 20 ms intervals [47].

### 2.12. In Vivo Bioluminescent Imaging

Bioluminescence from the sites of injections of a mixture of DNA-immunogen or vector DNA and pVaxLuc2 was measured on days 1, 2, 5, 7, 9, and 12 after the boost by in vivo imaging (Spectrum; Perkin Elmer, Waltham, MA, USA) as described previously [47,48]. Prior to capturing of the luminescent signal, mice were injected intraperitoneally with a solution of XenoLight D-Luciferin potassium salt (Perkin Elmer) in PBS at a dose of 150 µg/g body weight. Then, 10 min later, anesthesia was induced by 4% isoflurane and maintained by 2.5% isoflurane throughout the imaging procedure. Regions of interest (ROI) were localized around the injection sites, and the bioluminescence signal was quantified as the total photon flux (photons/s). Bioluminescence imaging data were processed using the Living Image^®^ software version 4.5 (Perkin Elmer).

### 2.13. End Point of DNA Immunization Experiment

DNA immunization experiment was terminated 2 weeks after the boost. Mice were bled through tail vein, blood was collected into 1.5 mL tubes, centrifugated, and sera were frozen at −20 °C for further antibody assays. After bleeding, mice were humanely euthanized by cervical dislocation, spleens were excised and homogenized, and single cell cultures were prepared using nylon 70 µm cell strainers (Nunc, Roskilde, Denmark). ACK Lysing Buffer (Thermo Fisher) was used to remove the erythrocytes. Stocks of murine splenocytes were prepared in RPMI containing 50% fetal calf serum and 10% DMSO, frozen at −80 °C for 1 week, and then transferred to liquid nitrogen for later assessment by flow cytometry.

### 2.14. Assessment of Cellular Immune Response

Cellular immune responses were assessed by multiparametric flow cytometry. Splenocytes of TERT-immunized and control vector-immunized mice were stimulated for 5 h in a CO_2_ incubator at 37 °C with solutions of TERT-derived peptides (Table 1; 10 µg/mL) or with a mix of mitogenic stimuli phorbol 12-myristate 13-acetate (PMA) (50 ng/mL) and ionomycin (1 µg/mL) (both from Sigma-Aldrich, St. Louis, MO, USA) in the presence of Golgi plug reagent (BD Pharmingen, Franklin Lakes, NJ, USA). After incubation, cells were stained for viability with the Fixable Viability Stain 660 (FVS660; BD Horizon #564405). Thereafter, cell surface staining was performed with a mixture of antibodies including FITC-conjugated anti-mouse CD8a (#553031) and APC-H7-conjugated anti-mouse CD4 (#560181). Cells were then washed, fixed, permeabilized using PerFix-nc Kit (Beckman Coulter, Brea, CA, USA), and stained with PE-conjugated anti-mouse IFN-γ antibodies (#557649), BV421-conjugated anti-mouse IL-2 antibodies (#562969), and BV510-conjugated anti-mouse TNF-α antibodies (#563386); all above antibodies were from BD Pharmingen. All stainings were performed in duplicates. Totally, five staining runs were done, each run including one sample from each of three groups. Stained samples were analyzed on a FACSAria II cytometer (BD Biosciences, Franklin Lakes, NJ, USA). Data were exported as FCS3.0 files using FACSuite software and analyzed using FlowJo X.07 program (FlowJo LLC, Ashland, DE, USA). First, general lymphocyte population was defined, and viable cells were identified by the lack of FSV660 staining. From the viable population, cells of interest were defined by the expression of CD4 and CD8 surface markers and for production of cytokines IFN-γ, IL-2, and TNF-α (Appendix A). Data were presented as percent of CD4+ or CD8+ cells producing one, two, or three cytokines, from the total population of CD4+ or CD8+ cells. Percent of cells positive for IFN-γ, IL-2, and TNF-α after stimulation with growth medium (background) was subtracted from all values.

### 2.15. Assessment of Anti-TERT Antibody Response

Plates (MaxiSorp, Sigma-Aldrich, St. Louis, MO, USA) were coated with TERT-derived peptides (Table 1) diluted in carbonate-bicarbonate buffer (pH 9.3) at concentration of 10 µg/mL by keeping overnight at room temperature and then incubated for 3 days at 6 °C or coated with rtTERT freshly diluted in PBS (pH 7.6) at concentration 0.3 µg/mL by keeping overnight at 6 °C. Plates were washed and blocked with PBS containing 10% normal goat serum, 2% bovine serum albumin, and 0.05% Tween 20 (Scan Buffer) for 1 h at room temperature. Blocking buffer was discarded prior to further assessment. Sera were diluted 1:100 with Scan Buffer and preincubated overnight at 6 °C. Then, they were further diluted with Scan Buffer in two- to threefold steps, distributed on pre-blocked plates, and incubated overnight at 6 °C. After that, plates were washed six times with PBS containing 0.1% Tween 20 and incubated for 1 h at 37 °C with goat anti-mouse IgG conjugated to HRP (Dako, Santa Clara, CA, USA) diluted in Scan Buffer. Thereafter, plates were washed as described above, dried by repeated tapping on filter paper, and incubated with 100 µL of freshly made 3,3′,5,5′-tetramethylbenzidine solution (Dako) for 10 min at room temperature. The reaction was stopped by adding 50 µL 0.1 M sulfuric acid, and plates were read on ELISA reader (Multiskan, Thermo Fisher, Waltham, MA, USA) at dual wave length of 450 and 620 nm. Optical density values were represented as OD450–OD620 (OD_450–620_). The end point dilution titer was established as serum dilution at which the OD_450–620_ of a well with immune mouse serum became equal to or lower than the average OD_450–620_ of 3 to 5 control mouse sera assessed on the same plate or ≤0.1.

### 2.16. Assessment of In Vivo Tumorigenicity of 4T1luc2 Clones Expressing rtTERT

The capacity of 4T1luc2 derivative clones expressing rtTERT to form tumors and metastases was tested by their ectopic implantation into 8-weeks-old female BALB/c mice done as was described previously [42]. Briefly, suspensions of parental or daughter rtTERT expressing 4T1luc2 clones in the doses of 2.5 × 10^3^, 5 × 10^3^, and 1 × 10^4^ cells in 50 μL of RPMI-1640 were injected into mice to the right and to the left of the base of the tail (*n* = 4 per dose per cell line). Injections were performed subcutaneously with a 25G needle mounted on an insulin syringe (B Braun, Melsungen, Germany). Tumor size was assessed by morphometric measurements done at regular intervals using calipers; tumor volume was calculated using standard formula for xenograft volume *V* = *xy*^2^⁄2. Cell growth was also assessed by bioluminescence imaging (BLI) as described above. Monitoring of bioluminescence was performed directly after the implantation, then on days 1, 2, 3, 5, and then every 2–3 days until the tumor volume of the first mouse in the experiment reached the volume of 1 cm^3^. Mice were weighed at each monitoring timepoint.

### 2.17. End Point of Tumor Challenge Experiment, Collection of Mouse Organs, and Rapid Ex Vivo Assessment of Metastases

When tumor volume of the first mouse in the experiment reached the volume of 1 cm^3^ (day 21 after implantation), mice were weighed and injected intraperitoneally with freshly prepared solution of XenoLight D-luciferin potassium salt (Perkin Elmer) in PBS in the amount of 150 µg/g based on the actual weight. After 8 min, mice were humanely euthanized. Tumors, lungs, liver, kidneys, and spleens were dissected with surgical scissors. Immediately after this, all organs were transferred into individual wells of 24-well black tissue culture test plates (Wallac, Turku, Finland) containing 2 mL RPMI-1640 medium and subjected to ex vivo BLI to assess the presence of Luc-expressing cells as described previously [42]. Thereafter, tumors, lungs, livers, kidneys, and 1/2 of the spleens were transferred into 5 mL of 4% formaldehyde solution in PBS, incubated for 24 to 48 h at 6 °C, then washed five times with PBS, and used to prepare FFPE blocks. The other half of the spleens was used in the evaluation of cellular immune response to TERT.

### 2.18. Evaluation of Cellular Response against TERT in Mice Implanted with 4T1luc2 Clones

The residual part of the spleens was homogenized and then, single cell cultures were prepared and frozen at −80 °C until the evaluation of cellular response against TERT by multiparametric flow cytometry. Cells were thawed at two occasions and assessed for the response against TERT1, TERT2, TERT6, and TERT8. Assessment was done for 4 mice per group (totally 12) in independent runs of 6 mice (two per group) each, as described above for DNA-immunized mice. PMA was used as a positive, and RPMI alone was used as a negative control. Percent of cells positive for IFN-γ, IL-2, and TNF-α after stimulation with growth medium (background) was subtracted from all values.

### 2.19. Tumor Histology and Ex Vivo Assessment of the Metastases

FFPE blocks were prepared from the formalin-fixed tumor tissues and organs. FFPE blocks were sectioned on a microtome according to the standard protocols (https://www.protocolsonline.com/histology/sample-preparation/paraffin-processing-of-tissue/). Sections mounted on glass slides were dewaxed, rehydrated, and stained with Mayer’s hematoxylin solution. Then, they were washed, rinsed, and counterstained with eosin Y solution, and after that, they were dehydrated, washed with absolute alcohol, and covered with cover slips for microscopic evaluation. The slides were examined by light microscopy (Leica DM500, Wetzlar, Germany). Histological evaluation of tumors was based on the standard parameters such as acinar formation, nucleus size and pleomorphism, and mitotic activity [49]. Grades of the tumors were calculated according to [50]. Lymphocytes infiltration was assessed in 15 high power (400×) microscope fields of hematoxylin–eosin-stained slides by computer-assisted morphometry using specialized NIS-Elements software (Nikon, Tokyo, Japan). Formalin-fixed, Paraplast-embedded liver tissues were used to diagnose and evaluate the formation of metastases. For each mouse, the area of tumor metastases was quantified in 15 high power (400×) microscope fields of hematoxylin–eosin-stained slides by computer-assisted morphometry using specialized NIS-Elements software (Nikon, Tokyo, Japan).

### 2.20. Statistical Analysis

Total photon flux from the site of coinjection of TERT DNA or TERT-HA DNA and Luc DNA was compared using ordinary two-way ANOVA with Dunnett’s multiple comparison test (GraphPad Prism 6, San Diego, CA, USA). Corrected total cell fluorescence (CTCF) was analyzed using ordinary one-way ANOVA with Holm-Sidak multiple comparison test (GraphPad Prism 6, Graphpad Software Inc., San Diego, CA, USA). Cell cycle analysis was analyzed using two- tailed *t*-test. Total photon flux from the site of injection of cells in tumorigenicity assessment was compared using RM two-way ANOVA with Dunnett’s multiple comparison test (GraphPad Prism 6). Relative percent of CD8+ and CD4+ T cells expressing one, two, or three cytokines after stimulation with each of TERT peptides, end point titers of mouse sera, and tumor volumes were compared using nonparametric Kruskal–Wallis and Mann–Whitney tests (Statistica AXA 11.0; TIBCO Software Limited, London, UK). Total photon flux values were correlated to relative percent of CD8+ and CD4+ T cells expressing one, two, or three cytokines after stimulation with each of TERT peptides. Relative *rtTERT* mRNA expression was correlated to the amount of *rtTERT* locus insertions in genome. Correlations were performed using Spearman ranking test (Statistica AXA 11.0). *p* values < 0.05 were considered significant.

## 3. Results

### 3.1. Design and Expression of Synthetic Rat TERT Gene

TERT of different species is a highly conserved protein with four major structural/functional domains: N-terminal (TEN), RNA-binding (TRBD), the reverse transcriptase (RT) domains, and the C-terminal extension (CTE) (Figure 1A). RT domain, bearing homology to other reverse transcriptases, is the enzyme which builds telomeres, whereas other domains are involved in RNA interaction and protein stabilization [51]. Rat TERT was chosen as the immunogen as it differs in sequence from TERT of mice and humans (UniProtKB, rat TERT Q673L6; mouse TERT O70372; human TERT O14746; Figure 1B) and could, therefore, be expected to overcome immune tolerance in these species in the stage of preclinical and clinical trials. Numerous amino acid mismatches between TERT variants could be seen in peptides encompassing clusters of T- and B-cell epitopes shown to be recognized by the murine and human immune system (Figure 1C, Table 1, and references therein).

Coding sequence of TERT optimized for expression in mammalian cells was designed and synthesized. We also designed TERT variant with a C-terminal hemagglutinin tag (TERT-HA) to enhance the antigenic dissimilarity (GenBank submissions MK749423 and MK749424). Synthetic genes were cloned into plasmid vector for DNA immunization pVAX1. Expression of the synthetic genes was tested in 293T cells. Staining with commercial rabbit polyclonal anti-TERT antibodies revealed weak bands corresponding to a protein with expected molecular mass of approximately 125 kDa (Appendix A). The level of expression was low, which supports earlier observations of high proteolytic instability of TERT, shown to be efficiently degraded by the proteasome [56,57,58].

### 3.2. TERT Epitope Map and Selection of TERT-Derived Peptides for the Analysis of Anti-TERT Immune Response

TERT-derived peptides were selected based on the published epitope data [30,31,32,33,34,35,36,37,38,39] and predictions of processing and presentation of TERT epitopes in the context of MHC class I and II, done using web tools of the Immune Epitope Database IEDB (www.iedb). Regions found to contain clusters of known B- and T cell epitopes are presented in Figure 1C–G. Peptides from these regions were selected, which had high probability of recognition by the immune system of BALB/c mice as containing cytotoxic T-cell (MHC-I Binding Predictions tool generating “Class I scores”; restriction to H2 Db and H2 Dd) and/or T-helper cell and B-cell epitopes (MHC-II Binding Predictions tool generating “Class II scores”; http://tools.iedb.org/mhcii/) (Figure 1C–G and Table 1).

### 3.3. Immunization and In Vivo Assessment of the Development of Immune Response by Bioluminescent Imaging

We used a method to assess the integral immune response against TERT which we named “antigen challenge” [47]. Specifically, we primed mice with TERT DNA or TERT-HA DNA, and then boosted with the same plasmids mixed with DNA encoding firefly luciferase (Luc DNA) (Figure 2A). We have earlier shown that immune response induced by DNA immunogen in prime eliminates cells coexpressing DNA immunogen and Luc after the boost with plasmid mixture. This results in a rapid loss of bioluminescence signal from the sites of booster injections in DNA-immunized mice compared to control mice [47]. Here, boosting of TERT or TERT-HA DNA-immunized mice with plasmids encoding TERT/TERT-HA and Luc resulted in a significant loss of bioluminescence signal in comparison to that in the control mice primed with pVAX1 and boosted with pVAX1 mixed with Luc DNA (*p* < 0.05; Figure 2B,C). Dynamics of the loss of photon flux in TERT- and TERT-HA-immunized groups were similar (*p* > 0.5; Figure 2B,C). Rapid loss of TERT/luciferase-expressing cells indicated that TERT- and TERT-HA DNA-immunization induced an immune response able to eliminate cells coexpressing TERT (TERT-HA) and Luc from the sites of immunization.

### 3.4. Assessment of Cellular Immune Response

Recognition by murine CD4+ and CD8+ T cells of the panel of TERT-derived peptides selected as potentially immunogenic was assessed at the experimental end point (Table 1, Figure 1C–G, and Appendix A). Data were presented as percentage of CD4+ and CD8+ T cells reacting to stimulation with TERT peptides by production of one, two, or three cytokines (Figure 3). CD4+ T cells specifically recognized peptides TERT1, 3, 5–8 and CD8+ T cells, TERT1, 5–8 (Figure 3A–C). Peptide TERT4 was not recognized, and TERT2 and TERT9 stimulated cytokine production in both TERT/TERT-HA and vector-immunized mice, i.e., contained autoepitopes of TERT (Figure 3A–C). Recognition was manifested by simultaneous production of IFN-γ/IL-2, IFN-γ/TNF-α, or IFN-γ/IL-2/TNF-α by CD4+ (Figure 3A,C) and CD8+ T cells (Figure 3B,C). Specific production of IL-2/TNF-α combination by CD4+ or CD8+ T cells was not detected (data not shown). Production of monocytokines by CD4+ and CD8+ T cells was mostly unspecific (Appendix A). Both TERT and TERT-HA DNA-immunized mice developed specific multicytokine CD4+ and CD8+ T cell response, significantly exceeding the response levels in the control animals (Figure 3A–C). At the same time, TERT/TERT-HA DNA and vector-immunized mice did not differ in their response to stimulation with mitogen PMA (Figure 3D). There was no difference in the spectrum or magnitude of responses in mice receiving TERT and TERT-HA DNA (Figure 3A,B), indicating that these two groups can be considered as one group of mice DNA-immunized with TERT.

We further looked whether recognition of TERT peptides was related to the difference between amino acid sequences of rat and murine TERT. Peptide TERT5 contains 8, TERT1 and TERT7—3, TERT2, TERT3, and TERT8—2, and TERT6—just 1 amino acid mismatch and were all able to stimulate CD4+ and/or CD8+ T cell response in immunized mice. TERT4 sequence in mice and rats is identical, and the peptide was not immunogenic. At the same time, TERT9, also identical in rats and mice, was recognized by CD4+ and CD8+ T cells of both immunized and control animals, indicating presence of an autoepitope (Figure 1G, Table 1, Figure 3A–C, and Appendix A). Another autoepitope recognized by control mice colocalized with TERT2, contained two amino acid mismatches between rat and murine TERT2 sequences. (Figure 1D, Table 1, Figure 3A–C, and Appendix A). The immune recognition of both TERT2 and TERT9 reflected an immune response to endogenous TERT, not rat TERT introduced by DNA immunization. On the overall, the degree of amino acid dissimilarity between TERT sequences in the immunogen and in the host was not translated into stronger or weaker specific immune recognition.

Further, we looked into the immune recognition of peptides which have induced specific immune stimulation in >0.1% of CD4+ and CD8+ T cells (Figure 3A–C) in individual TERT and TERT-HA DNA-immunized mice. This limited the peptide panel to TERT1 and TERT6-8 (Figure 4A,B). Results were presented as pile-up of populations of CD4+ and CD8+ T cells which responded to peptide stimulation by production of just one, or two, or all three cytokines (nonoverlapping populations). Analysis of the profiles of cellular immune response in mice immunized with TERT/TERT-HA DNA demonstrated that TERT1, TERT6, TERT7, and TERT8 harbored epitopes of CD4+ and CD8+ T cells capable of multiple cytokine secretion, with dominance of IFN-γ (Figure 4A,B). To check the functionality of this response, we looked for the correlations between percent of CD4+ and CD8+ T cells producing IFN-γ, IL-2, and/or TNF-γ in different combinations and loss of bioluminescence from the sites of TERT (TERT-HA)/Luc DNA boosts. Residual percent of bioluminescence left at immunization sites by days 9 and 12 after the boost inversely correlated with the magnitude of multicytokine response of CD4+ T cells to stimulation with TERT6, TERT7, and TERT8 and of CD8+ T cells, to stimulation with TERT6 and TERT8 (*p* < 0.01; Appendix A). Correlation of the loss of bioluminescence with cellular response to TERT1 and TERT5 was of less significance and involved only IFN-γ /IL-2/TNF-γ producing CD8+ T cells (day 12; Appendix A). These correlations pointed at the lytic potential of CD8+ and CD4+ T cells specific to the epitopes localized between aa 845–929 of TERT, i.e., in the RT domain.

### 3.5. Antibody Response

TERT and TERT-HA DNA induced an antibody response recognizing recombinant rtTERT and two peptides from this region, TERT7 and TERT 8, with IgG titers in the range of 2–7 × 10^3^ (Figure 5). Other TERT peptides were not recognized (Figure 5). Thus, B-cell response induced by TERT DNA immunization was targeted against aa 888–929 of TERT, within the region containing a cluster of epitopes of CD4+ and CD8+ T cells.

With this, we have shown that immunization with TERT DNA induces cellular and antibody response targeting the region between aa 845 and 929 localized within RT domain of TERT (rtTERT), and this response correlates to in vivo clearance of TERT/reporter coexpressing cells. This positions rtTERT as a necessary and sufficient component of cancer vaccine.

### 3.6. Generation of 4T1luc2 Clones Expressing RT Domain of Rat TERT

Earlier, we found that expression of HIV-1 RT by murine mammary adenocarcinoma 4T1luc2 cells enhances their tumorigenic and metastatic activity [42]. We inquired if this could be the case for RT domain of TERT, which, in this case, will be unsafe to use as a vaccine component. To test this possibility, we generated variants of 4T1luc2 cells expressing rtTERT and evaluated their capacity to form tumors and metastasis. Transduction of 4T1luc2 cells with lentivirus encoding rtTERT yielded four daughter clones with genomic inserts of *rtTERT* (i.e., 4T1luc2_rtTERT_C6, 4T1luc2_rtTERT_B5, 4T1luc2_rtTERT_H9, and 4T1luc2_rtTERT_F1; Table 2).

Number of inserts of *rtTERT* into 4T1luc2 genome was assessed using ddPCR and compared with the copy number of two housekeeping genes, beta-actin (*Actb*) and myostatin (*Mstn*), both with ploidy 1 (two copies per genome). For three cell lines, *Mstn/Actb* relative copy number was close to 2, and for 4T1luc2_rtTERT_H9 line, it was close to 3.5, which implies a change in the relative copy number of these house-keeping genes towards each other (Table 2 and Appendix A). Level of expression of *rtTERT* mRNA by the generated clones was assessed using real-time PCR, calculated by 2-ddCt method and normalized to the level of *rtTERT* mRNA in the 4T1luc2_rtTERT_B5 cell line where it was the lowest (Table 2). All clones demonstrated detectable levels of expression of *rtTERT* mRNA (Table 2) which tended to correlate to the copy number of *rtTERT* (Spearman test, r = 1, *p* = 0.08, *n* = 4).

We performed cell cycle analysis of the daughter clones compared to the parental 4T1luc2 cells. The analysis revealed that three daughter clones had reduced population of cells in G0/G1 and increased population of cells in S and G2/M phase areas (Figure 6C) indicative of increased cell proliferation. At the same time, the doubling time for all subclones was similar to that of the parental cells over 3 weeks of consequent observations (Kruskal–Wallis, *p* > 0.05, *n* = 10).

Observed changes in the cell cycle prompted us to evaluate the genomic stability of rtTERT-expressing 4T1luc2 subclones. The assay was done using immunofluorescent microscopy with antibodies detecting phosphorylated form of H2A histone family member X, which forms in response to double-stranded DNA breaks (γ-H2AX foci; [59]). Interestingly, highly malignant tumor cell line 4T1luc2, genetically unstable, p53^null^, with complex aneuploid karyotype [60,61,62], and two of the subclones, 4T1luc2_rtTERT_C6 and 4T1luc2_rtTERT_H9, appeared to be relatively genomically stable (few γ-H2AX foci; Figure 7A, panel I, and Figure 7A, panel II,III,E). On contrary, 4T1luc2_rtTERT_F1 and 4T1luc2_rtTERT_B5 exhibited intense disperse anti-γ-H2AX nucleus staining (Figure 7A, panel IV,V,E) indicative of double-stranded DNA breaks, a hallmark of genomic instability.

High genomic instability can induce overexpression of the endogenous TERT via multiple nonexclusive mechanisms [63,64]. We analyzed if this was the case for the clones overexpressing rtTERT. The level of expression of endogenous TERT was assessed by immunofluorescence staining of cells with anti-TERT antibodies against an epitope outside its reverse transcriptase domain (Ab 191523), i.e., not detecting the expression of rtTERT introduced by lentiviral transduction (Figure 7A, panels I–V). Series of simultaneous daughter and parental cell line stainings with Ab191523 were performed, immunofluorescence signals of single cells were quantified, and the average ratio of anti-TERT signal generated by daughter clones to that of the parental 4T1luc2 cells was calculated. This revealed a significant increase in the expression of endogenous TERT by 4T1luc2_rtTERT_F1 and 4T1luc2_rtTERT_B5 cells (which exhibited genomic instability), whereas the levels of expression of endogenous TERT by two other clones did not differ from that in 4T1luc2 cells (Figure 7A, panels IV, V versus panels II, III, compared to I, respectively; Figure 7A).

Thus, among four generated clones, two were significantly different from the parental 4T1luc2 cells in their cell cycle characteristics, the degree of genomic instability, and levels of expression of endogenous TERT. These made them inapplicable for the study of the effect(s) of rtTERT expression on the capacity of 4T1luc2 cells to form tumors and metastasis. Of note, both B5 and F1clones were characterized by the low number of *rtTERT* genomic inserts and low levels of expression of rtTERT (Table 2) indicating that the observed alterations were unrelated to overexpression of exogenous rtTERT. Two remaining clones 4T1luc2_rtTERT_C6 and 4T1luc2_rtTERT_H9, similar to 4T1luc2 in genomic stability, cell cycle parameters, and levels of expression of endogenous TERT, were taken for further in vivo assessment of tumorigenicity.

### 3.7. Tumorigenic Potential of 4T1luc2 Clones Expressing RT-TERT

Clones 4T1luc2_rtTERT_C6, 4T1luc2_rtTERT_H9, and parental 4T1luc2 cells were ectopically implanted into 8-week-old female BALB/c mice in three different doses, and tumor formation was monitored by in vivo BLI and calipering (Figure 8). After injection of 2.5 × 10^3^ cells, rtTERT-expressing cell lines either formed small tumors, which grew at a slower rate than the parental clones (clone H9), or rejected tumors without growth (clone C6) (Figure 8A). Injections of 5 × 10^3^ and 1 × 10^4^ 4T1luc2_rtTERT_H9 cells led to the formation of slowly growing tumors, which by experimental end point were 3–5 times smaller than tumors formed by the parental clone (Figure 8B–E). Of eight injections of 5 × 10^3^ or 1 × 10^4^ 4T1luc2_rtTERT_C6 cells, six were rejected after transient growth or without growth (Figure 8B–E). Mice that rejected tumors had scars at the site of injection (after healing of inflammatory infiltrates) sized <1 mm^3^.

Tumors were excised, fixed, paraffin embedded, sectioned, stained with hematoxylin and eosin, and subjected to the analysis of morphology with tumor grading (Figure 8F–H). All tumors could be characterized as high grade (G3) tumors of low differentiation with mixed epithelioid (dominant) and sarcomatoid appearance. Multitude of cells within tumors had pleomorphic nuclei with several nucleoli. For 4T1luc2 and 4T1luc2_rtTERT_C6, the mitotic activity was low to moderate, whereas for tumors formed by 4T1luc2_rtTERT_H9, the mitotic activity was high. Atypical mitosis was presented in all samples. Tumors had multiple areas of multifocal necrosis surrounded by the inflammatory infiltrates consisting mostly of neutrophils and mononuclear cells; non-necrotic intratumoral tissues contained single tumor cells. Tumors demonstrated invasive growth into muscles and subcutaneous fat (Figure 8F–H). Altogether, morphologically, tumors were undistinguishable from those formed by the parental clone, i.e., expression of rtTERT limited the in vivo tumor growth, but not the characteristics of the tumors formed by 4T1luc2 clones.

### 3.8. Metastatic Potential of 4T1luc2 Clones Expressing rtTERT

Further, we compared the metastatic activity of 4T1luc2 and rtTERT-expressing daughter clones. For this, we assessed organs of mice excised at the experimental end point for the presence of Luc-expressing tumor cells using ex vivo imaging [42]. Specifically, we monitored total photon flux from lungs, liver, spleen, and kidneys of mice implanted with 4T1luc2_rtTERT_C6 and 4T1luc2_rtTERT_H9 as compared to the parental 4T1luc2 cells. Photon flux from lungs and livers of mice bearing rtTERT-expressing tumors was significantly lower than signal emitted by lungs and livers of mice implanted with 4T1luc2 (Figure 9A,B, respectively). Photon flux from other organs of 4T1luc2_rtTERT_C6- and 4T1luc2_rtTERT_H9-implanted mice did not exceed the background levels indicating that these organs, on contrary to the organs of 4T1luc2-implanted mice, were not infiltrated by Luc-expressing tumor cells (Figure 9C,D).

Liver appears to be a highly representative organ for the histological assessment of metastatic activity of 4T1luc2 cells [42,65]. Although 4T1luc2 cells poorly metastasize into the liver, more aggressive 4T1luc2 variants readily form liver metastases [42]. With this in mind, we performed the histochemical study of the livers of mice implanted with 4T1luc2 cells expressing rtTERT. Livers were excised, fixed, paraffin embedded, sectioned, stained with hematoxylin and eosin, and analyzed for the presence of metastases, single infiltrating tumor cells, and leukocytes (Figure 9E–G). Cell lines 4T1luc2_rtTERT_H9 and 4T1luc2_rtTERT_C6 generated significantly lower average number of liver metastases than the parental cell line (*p* < 0.001; Figure 9H). The average size of liver metastases did not differ (Figure 9I). Number of liver metastases correlated with the photon flux (max radiance) from the liver, reflecting the number of Luc-expressing tumor cells infiltrating the liver (R = 0.5; *p* = 0.024; Appendix A). The latter demonstrated the usefulness of ex vivo organ imaging as a rapid high-throughput method of evaluation of metastatic activity.

We have also assessed leukocyte infiltration into the liver. Implantation of 4T1luc2_rtTERT_C6 induced significantly lower leukocyte infiltration into the liver than implantation of 4T1luc2 (*p* < 0.01); for 4T1luc2_rtTERT_H9 cells, the effect was similar to that of the parental cells (Figure 9J; *p* > 0.05). Leukocyte infiltration into the liver correlated with the number of liver metastases (*p* = 0.00001; Appendix A), indicating that infiltrating tumor cells attracted cells of the immune system. Thus, we confirmed that rtTERT expressing 4T1luc2 clones were compromised in their capacity to grow in vivo and form metastasis in the distal organs.

### 3.9. Cellular Immune Response against Epitopes of TERT in Mice Implanted with rtTERT-Expressing 4T1luc2 Cells

Finally, we assessed if limited growth of rtTERT-expressing tumors was related to an adaptive immune response against TERT/rtTERT. For this, we assessed immune response to TERT by splenocytes of representative number of mice from each of the groups, slitting them into those rejecting tumors (*n* = 4), restricting tumor growth (*n* = 4), and not restricting tumor growth (*n* = 4). Mice were assessed in two independent runs, taking two mice from each cluster. Their splenocytes were tested for the capacity to produce IFN-γ, IL-2, and TNF-α in response to stimulation with TERT6 and TERT8, which we have shown to be specifically recognized by T cells of mice DNA-immunized with rat TERT (TERT7 was excluded as it partially overlapped TERT8) and against epitopes of TERT lying outside rtTERT: TERT1 as an epitope inducing T cell response in DNA-immunized mice and TERT2 as an autoepitope of TERT recognized by all mice. Cytokine production was assessed by multiparametric flow cytometry and presented as percent of mono-, di-, and tricytokine producing CD4+ and CD8+ T cells (as for DNA immunized mice). Stimulation with PMA served as a positive control. To get an integrative overview of T cell reactivity, we presented data as a pile up of all TERT-reactive T cell populations (Figure 10A,B).

Mice implanted with 4T1luc2_rtTERT_C6 line not developing tumors demonstrated no cytokine response to stimulation with any of the TERT-derived peptides (Figure 10A,B and Appendix A). Mice implanted with 4T1luc2-rtTERT_H9, capable to restrict tumor growth, exhibited increased percent of CD4+ and CD8+ T cells responding to stimulation with TERT1, 6, and 8 by mono- and multicytokine production, dominated by secretion of TNF-γ (up to 6% of CD4+ and 11% of CD8+ T cells) and also cytokine response to the autoepitope in TERT2 (Figure 10A,B and Appendix A). Mice implanted with the parental cell line, unable to limit tumor growth, responded to TERT1, 6, and 8 by limited cytokine production: reactive CD4+ constituted 1–2%, and CD8+—2–5% of respective T cell populations (Figure 10A,B). At the same time, their T cells strongly reacted to stimulation with TERT2: reactive CD4+ and reactive CD8+ constituted 6% and over 10% of the total CD4+ and CD8+ T cells, respectively (Figure 10A,B and Appendix A).

We analyzed the overall profile of cytokine response to TERT epitopes by the Friedman ANOVA test with Kendall Coefficient of concordance applying a stringent criterion of statistical difference (*p* value was set at <0.01 to detect only highly significant differences). Analysis demonstrated similarity in the overall profile of cytokine response to the autoepitope in TERT2 in all mice with growing tumors (implanted with 4T1luc2 or with 4T1luc2_rtTERT_H9), whereas mice that rejected challenge with 4T1luc2_rtTERT_C6 demonstrated no response to TERT2 by either CD4+ or CD8+ T cells (Figure 10A,B; Appendix A). There was no difference in response to TERT2 in mice able (4T1luc2_rtTERT_H9) and unable (4T1luc2) to restrict tumor growth (*p* > 0.05; Appendix A). This indicated that T cell immune response against the autoepitope in TERT2 was not involved in the immune control of tumor growth. TERT1 induced strong TNF-α production, mainly by CD8+ T cells, in mice that restricted growth of 4T1luc2_rtTERT_H9 tumors, low response in mice with unrestricted growth of 4T1luc2 tumors, and no response in mice rejecting challenge with 4T1luc2_rtTERT_C6 cells (Figure 10A,B and Appendix A). Major difference was noted in the T cell response to stimulation with TERT6 and TERT8, presented by rtTERT-expressing 4T1luc2 cells. TERT6 and TERT8 induced strong multicytokine response of CD4+ and CD8+ cells in mice that restricted growth of 4T1luc2_rtTERT_H9 tumors, low response in mice with unrestricted growth of 4T1luc2 tumors, and no response in mice rejecting 4T1luc2_rtTERT_C6 cells (Figure 10A,B; Appendix A). Assessment of the overall profile of cytokine production in response to TERT6 and TERT8 allowed to discriminate mice with growing tumors (4T1luc2) from those restricting tumor growth (4T1luc2_rtTERT_H9) and those rejecting tumors (4T1luc2_rtTERT_C6) (Appendix A). This phenomenon was not related to the overall reactivity of T cells, as 4T1luc2-, 4T1luc2_rtTERT_C6-, and 4T1luc2_rtTERT_H9-implanted mice did not differ in their response to stimulation with mitogen PMA (Appendix A).

Further, we assessed the relation between growth of tumors in mice implanted with 4T1luc2 clones (by tumor size at the experimental end point) and specific parameters of cellular immune response against autoepitope TERT2 and epitopes represented by TERT1, 6, and 8 on the example of IFN-γ/IL-2 secreting CD4+ and CD8+ T cells. As was shown for the overall profile of cytokine production by CD4+ and CD8+ T cells (Appendix A), mice not restricting or partially restricting tumor growth did not differ in the percent of TERT2-specific IFN-γ/IL-2 secreting CD4+ or CD8+ T cells (Appendix A). At the same time, mice partially suppressing tumor growth tended to have higher percent of IFN-γ/IL-2-positive CD8+ responding to TERT1, 6, and 8; difference for CD4+ T cells did not reach the level of significance (Appendix A). Mice that rejected the tumors had no TERT-specific CD4+ or CD8+ T cells (Appendix A).

Next, we correlated the parameters of CD4+ and CD8+ T cell response to TERT1, 2, 6, 8, to tumor growth. Percent of CD4+ or CD8+ T cell specific to TERT1, TERT6, or TERT8 did not correlate with either photon flux from tumors by day 16 or tumor size by the experimental end point (all *p* values >0.05; data not shown). At the same time, tumor size tended to correlate to the percent of TERT2-specific CD4+ T cells responding TERT2 stimulation by production of IL-2, IFN-g/IL-2, and IFN-g/IL-2/TNF-a (p values in the range of 0.058–0.071), whereas no correlation was observed for TERT2-specific CD8+ T cells (Appendix A). Furthermore, for mice implanted with 4T1luc2_rtTERT_C6 and 4T1luc2_rtTERT_H9 cells, photon flux from the sites of injections positively correlated with percent of CD4+ T cells responding to TERT2 with production of TNF-a, and tended to correlate with percent of TERT2-specific IFN-γ/TNF-γ- and IFN-γ/IL-2/TNF-γ-positive CD4+ T cells (Appendix A). Statistical significance in correlations was difficult to reach since all mice that rejected tumors without growth or after transient growth had no tumors, only measurable were the scars entered as formations with the size 0.5 to 1 mm^3^.

Thus, mice with growing tumors developed strong multicytokine response to autoepitope in TERT2, which was positively correlated to tumor growth. Capacity to build multicytokine CD4+ and CD8+ T cell response against epitopes in TERT1, TERT6, and TERT8 (specifically, TERT6 and TERT8, Appendix A) discriminated mice that were able to restrict tumor growth. The response was elicited mainly by IFN-γ/IL-2-secreting CD8+ T cells (Appendix A). However, magnitude of the response expressed as populations of IFN-γ/IL-2/TNF-γ producing TERT6- and TERT8-specific CD4+ and CD8+ T cells showed no inverse correlation to the size of the rtTERT-expressing tumors (i.e., it was not a decisive factor in tumor rejection). Finally, mice that completely suppressed tumor growth demonstrated no T cell response to TERT, either autoepitope TERT2 or CD4+ and CD8+ T cell epitopes represented by TERT1, TERT6, and TERT8, i.e., the rejection was not dependent on the adaptive immune response. The latter phenomenon needs further study to define the underlying mechanism(s), which act in vivo, as cell line in question, 4T1luc2_rtTERT_C6, demonstrated in vitro growth parameters undistinguishable from 4T1luc2 and 4T1luc2_rtTERT_H9.

## 4. Discussion

Cancer vaccines of today widely employ nucleic acids, as naked DNA, RNA, or viral vectors. Vaccines based on TERT, a classical TAA overexpressed in the majority of tumors, are not an exclusion: four are based on plasmid DNA (NCT00753415, NCT02960594, NCT03265717, and NCT03502785). The best were shown to induce notable immune response in clinical trials [28,29]. However, fine specificity of anti-TERT immune response and immune correlates of their antitumor activity remain largely unknown. Here, we have developed and tested a new DNA vaccine targeting TERT based on the amino acid sequence of rat TERT, expected to overcome potential tolerance in preclinical trials in mice and potential clinical trials. Rat TERT and its variant with the C-terminal HA-tag encoded by synthetic expression-optimized genes, were used to immunize mice using an optimized protocol consisting of repeated intradermal injections followed by electroporation [47]. The effector potential of integral anti-TERT immune response was assessed in the “antigen challenge” experiment. Mice were primed with TERT or TERT-HA DNA and then boosted with the same plasmids mixed with DNA encoding luciferase (Luc); bioluminescence from the sites of booster injections was followed by in vivo bioluminescent imaging. Mice DNA-immunized with TERT/Luc demonstrated rapid disappearance of bioluminescence compared to controls boosted with vector/Luc, indicating presence of an effector immune response against TERT capable to clear TERT/Luc-coexpressing cells from the sites of immunization.

Next, we analyzed specificity of anti-TERT immune response, identifying types of the involved T cells and mapping their epitopes. For this, we assembled an epitopic map of TERT and localized regions containing clusters of T cell epitopes recognized by cancer patients and by animals (mainly mice) in the preclinical vaccine trials (Figure 1). These regions were represented by synthetic peptides (TERT1 to TERT9; Table 1). Assessment of CD4+ and CD8+ T cells of immunized mice for the capacity to produce IFN-g, IL-2, and TNF-a in response to stimulation with TERT peptides done by multiparametric flow cytometry demonstrated that out of nine peptides, eight (all except TERT4) contained T cell epitopes recognized by the immune system of BALB/c mice. TERT1, TERT3, and TERT5–8 were recognized by mice DNA immunized with TERT/TERT-HA, whereas TERT2 and TERT9 induced potent cytokine secretion in all tested animals, including vector-immunized controls, i.e., contained autoepitopes of TERT. Specific immune response was manifested by coproduction of IFN-γ/IL-2/TNF-α characteristic to the effector CD8+ and CD4+ T cells, which can eliminate tumor cells [66]. Importantly, loss of bioluminescence in the “antigen challenge” test was proportional to the percent of CD4+ and CD8+ T cells responding to stimulation with TERT6, TERT7, and TERT8 by simultaneous production of two or three cytokines. All three localized at aa 845–929 of TERT within the RT domain of TERT (rtTERT). HA-tag had no effect on the magnitude or profile of TERT-specific immune response. TERT DNA-immunized mice also developed an antibody response against TERT7 and TERT8 and recombinant rtTERT (but not other TERT-derived peptides). Earlier, a study of therapeutic TERT vaccine (viral vector in prime, DNA in boost) demonstrated its efficacy in treatment of canine lymphomas [67]. Its therapeutic effect correlated with the development of anti-TERT antibodies assessed by ELISA made using pools of TERT-derived peptides [67]. Our study indicates that these B-cell epitopes may be localized at aa 888–929 of TERT within rtTERT domain. Thus, we found the main immunogenic region of TERT to be localized in its RT domain, whereas immune response to other TERT regions was either autoimmune in nature or dispensable as not related to the immune clearance of TERT-expressing tumor cells.

This data led us to the hypothesis that DNA vaccine targeting TERT can be based on rtTERT and omit the rest of the protein. RT domain, located near the C-terminus of TERT (Figure 1), contains seven motifs (1, 2, A, B, C, D, and E) that are conserved in all RT families. These motives are located in the fingers and palm subdomains. They mediate specific interactions with the template, primer, and nucleotides and perform similar functions in all RTs, including RT of HIV-1, except for an insertion between motifs A and B’ within the fingers subdomain, which is involved in multiple repeat addition exerted solely by TERT [68]. As we have found earlier, expression by tumor cells of HIV-1 RT makes them to form large and fast-growing tumors, as well as enhances their metastatic activity in mice [42]. If this is a property of all RTs, it can make DNA immunization with rtTERT unsafe, specifically in case of promising intratumoral immunotherapy [69], rtTERT expression by rtTERT-based DNA vaccine may enhance tumor aggressiveness.

To delineate if this could be the case, we generated clones of murine mammary gland adenocarcinoma cell line 4T1luc2 expressing rtTERT and evaluated their in vitro properties and their potential to form tumors and metastasis upon implantation into syngenic BALB/c mice as compared to the parental cells. Of four 4T1luc2 clones made to express rtTERT, three had increased population of cells in S and G2/M phase areas indicative of enhanced cell proliferation, corroborating earlier findings of transgenic expression of TERT extending cell proliferation phase [70], a property which could thus be attributed to the expression of RT domain. Two of three clones with increased cell proliferation exhibited multiple foci formed by the phosphorylated form of H2A histone family member X (γ-H2AX foci; [59]), significantly exceeding their level in the parental cell line. Such foci may reflect genomic instability of the clones [59] due to lentiviral transduction. The other explanation could be formation of the foci over uncapped (shelterin unprotected) telomeres (telomer dysfunction induced-foci, TIFs) because they are overexpressed, or too short to allow protein binding, or both [71,72]. Such dysfunctional telomeres or telomer-like structures can be synthesized by rtTERT complemented with TERT RNA-binding domain (TRBD) of endogenous TERT in complex with the endogenous telomerase RNA template (TERC) abundant in cancer cells [73]. Of note, RT domain alone can extend DNA and RNA primers to form short dsDNA and RNA/DNA hybrids [73,74]. Defective telomeres are exposed to DNA damage surveillance and could be degraded and/or subjected to end-to-end fusion events resulting in the overall genomic instability [72,75]. Resulting genomic instability/DNA damage can recruit endogenous TERT increasing its expression [63,64,76,77,78,79]. Indeed, clones with multiple γ-H2AX foci demonstrated increased levels of endogenous TERT, however, this involved only clones with a low number of *rtTERT* genomic inserts (<1, Table 2). It is left to be found if (i) γ-H2AX foci reflect genomic instability of these specific clones after lentiviral transduction, causing overexpression of endogenous TERT or (ii) expression of exogenous rtTERT was the primary event leading to the synthesis of dsDNA and/or DNA/RNA hybrids and/or aberrant TERT activity due to complementation with endogenous TRBD and TERC resulting in the aberrant formation of uncapped telomeres, both causing formation of γ-H2AX foci/TIFs and (compensatory) overexpression of endogenous TERT. Mechanistic aspects of this model of ectopic expression of rtTERT deserve a separate study. In the current one, we concluded that the genomic instability and increase in expression of endogenous TERT made clones 4T1Luc2_rtTERT_F1 and 4T1Luc2_rtTERT_B5 unfit (inadequate) for a comparative in vivo tumorigenicity study. Study of the in vivo effects of expression of RT TERT on tumorigenicity of murine adenocarcinoma 4T1luc2 cells was, therefore, carried on cell lines 4T1luc2_rtTERT_C6 and 4T1luc2_rtTERT_H9.

In vivo study in syngenic BALB/c mice demonstrated that 4T1luc2_rtTERT_C6 cells were severely compromised and 4T1luc2_rtTERT_H9 cells restricted in the capacity to grow in vivo. Tumors either did not grow (4T1luc2_rtTERT_C6) or grew slower and were of smaller size than those formed by the parental cells. Thus, on contrary to the effect of HIV-RT [42], expression by 4T1luc2 cells of RT domain of TERT did not lead to increase of their tumorigenicity, but instead restricted or even prohibited tumor growth. Metastatic activity of rtTERT-expressing 4T1luc2 cells was evaluated by ex vivo BLI of murine organs revealing presence of tumor cells. Tumor cells disseminated mainly into the lungs, whereas tumor cells in other organs were either very few or not detected. Ex vivo imaging data were corroborated by histochemical assessment done for the liver to compare with the data generated for HIV RT-expressing 4T1luc2 cells [42]. Importantly, maximum radiance from the liver correlated with the number of liver metastasis (R = 0.5; *p* = 0.024; Appendix A), supporting the utility of BLI of excised organs as a high-throughput method to detect and quantify the metastasis. Altogether, data obtained by ex vivo bioluminescent organ imaging and selective histochemical assessment demonstrated that rtTERT-expressing 4T1luc2 clones were severely compromised in the metastatic activity compared to 4T1luc2 cells expressing HIV-1 RT [42], to the ancestral 4T1 cells and even to the parental 4T1luc2 cells (here and [65]).

Finally, at the experimental end point, we assessed anti-TERT immune response induced in mice injected with 4T1luc2 subclones. Mice restricting tumor growth were characterized by high percent of CD4+ and CD8+ T cells responding to TERT6 and TERT8 by multicytokine production (Appendix A). Such cells were low or absent in mice bearing adenocarcinomas formed by 4T1luc2 cells (Appendix A). Multicytokine response against these epitopes in rtTERT was dominated by the production of TNF-γ. In DNA-immunized mice, such response correlated with the loss of TERT/Luc-coexpressing cells from the sites of immunization. However, here, it did not show any correlation with the tumor size (Appendix A). Furthermore, mice rejecting tumors exhibited no CD4+ or CD8+ T response to TERT1, TERT6, or TERT8, indicating that T cell response against these epitopes was not a critical determinant of the restricted tumor growth.

Earlier studies described the rejection of tumor cells after transient growth or without growth by neutrophils and macrophages (not tumor-associated macrophages) [80]. In this and other cases, rejection depended on the local availability (secretion or introduction) of multiple cytokines (IFN-γ, IL-2, IL-4, IL-7, IL-12, TNF-γ, GM CSF, and IFN-γ/β) [80,81,82,83,84,85,86,87]. Much attention had been drawn to type I IFNs constituting the first line of defense against cancer cells [88,89]. Type I IFNs are expressed in response to intracellular sensing of DNA, RNA, and DNA/RNA hybrids by the sensor proteins, such as Activation of the STimulator of INterferon Genes (STING), Retinoic acid-inducible gene-I (RIG-I), and Toll Like Receptors 7, 8, and 9 [90,91]. In transplanted tumor models in mice, systemic activation of DNA and RNA sensors triggers potent production of IFNα/β resulting in the death of tumor cells [92,93,94,95]. DNA sensing in cancer cells may be disrupted, but they retain a functional RNA-sensing mechanism [96]. We hypothesize that expression of rtTERT could have promoted this type of response through rtTERT-mediated expression of telomeric repeat-containing intracellular, shorter, and more stable cell-free RNA (TERRA and cfTERRA) [97,98] and/or telomeric DNA/RNA hybrids as regular (by)products of TERT activity [99]. TERRA, inside secreted exosomes, had been shown to mediate short-range innate immune signaling [97,98]. DNA/RNA hybrids are also effective in triggering type I IFNs [95]. Their formation detected by nucleic acid sensors could switch production of type I IFNs, which would restrict the growth of rtTERT-expressing tumor cells.

In both 4T1luc2- and 4T1luc2_rtTERT_H9-implanted mice developing tumors, we observed potent autoimmune response against TERT2, an autoepitope localized outside rtTERT. Percent of IFN-γ/IL-2 secreting CD4+ T cells specific to this autoepitope positively correlated with the tumor size. No such correlation was observed for CD8+ T cells, indicating that the magnitude of immune response, at least by CD8+ T cells, was not straightforwardly determined by the tumor size and availability of endogenous TERT as immunogen. Of note, in DNA immunization, clearance of TERT/Luc-coexpressing cells was not correlated to the percent of CD4+ or CD8+ T cells specific to TERT2. Altogether, these data pointed at inability of TERT2-specific T cells to eradicate TERT-expressing (tumor) cells and limit tumor growth and even indicated a promotion of tumor growth by TERT2-specific CD4+ cells. This is in line with the earlier studies in BALB/c mice, which have shown that depletion of CD4+ T cells specific to TAA causes regression of syngenic tumors [100]. This observation is highly relevant to human cancer, as it was earlier shown that strong anti-TERT response by CD4+ T cells manifested by the production of IFN-γ/IL-2 supports aggressive tumor growth [101].

In the context of cancer, CD4+ T cell subpopulations, immunosuppressive regulatory (Tregs), and pro-inflammatory T helper 17 (Th17) cells, which normally act to fine-balance the adaptive immune responses, promote cancer progression and metastasis. Th17 cells shape tumor microenvironment, defining tumor phenotypes; Tregs promote metastasis and metastatic tumor foci development [102]. Recognition of certain TAA autoepitopes by T cells lacking lytic potential, particularly by Tregs and/or Th17 CD4+ T cells, may lead to their infiltration into tumors and mouse organs harboring tumor cells, with subsequent stimulation of tumor growth and metastases formation. Indeed, we observed a correlation between the numbers of organ infiltrating leukocytes and resident tumor cells. Presence of such CD4+ epitope(s) in a cancer vaccine is highly undesirable. Vaccine candidate(s) are preferable that could suppress an autoimmune inflammatory response in favor of a lytic response against epitopes, preferably of CD8+ T cells, not preexisting, but induced by immunization. Interestingly, mice which completely suppressed or restricted tumor growth exhibited no TERT2-specific autoimmune response, registered in all control animals. Disappearance of this response may indicate induction of tolerance (depletion of reactive clones from T cell repertoire). Possibility of such retargeting of the immune response away from the decoy (auto) epitopes, which induce inflammation and promote tumor growth, requests further study.

Thus, we have shown that DNA immunization with rat TERT induces a specific cellular and antibody response against its RT domain with a lytic potential, which may be harnessed to control tumor growth. On the contrary to HIV-1 RT, expression of rtTERT does not enhance tumorigenic properties of cancer cells (4T1luc2), but instead reduces their capacity to form tumors and metastasis, induces CD8+ T cell immune response to epitopes within rtTERT associated with restriction of tumor growth, and suppresses CD4+ T cell response to an autoepitope of TERT associated with tumor growth and metastatic activity. This indicates that TERT vaccine based on its RT domain would be safe to use, even for intratumoral immunization.

This raises a question whether cancer vaccine should in fact be based on the full length TERT, or rtTERT domain may be sufficient? Use of a single domain may provide certain benefits: (a) focusing of the immune response on the epitopes of lytic CD8+ and CD4+ T cells and reducing the immune response, which has no lytic potential and is not protective; (b) excluding autoepitopes as decoy epitopes with specific T cell immune response potentiating tumor growth; and (c) reducing size of DNA vaccine, which would result in a more efficacious DNA delivery [103,104].

Another aspect worth mentioning is the safety issue of TERT DNA vaccination. Deletion of the catalytic VDD triplet (and in some studies, of the nucleolar localization sequence) is considered sufficient to make TERT safe as DNA vaccine [30,105]. However, it was shown that the effect of TERT on cell proliferation is only in part linked to its enzymatic activity, as inactivated TERT can still stimulate cell proliferation [106,107]. TERT enzymatic activity is not a sole factor determining TERT-driven cell immortalization. TERT is a multifunctional protein capable to regulate gene expression, modulate cell signaling, influence cell cycle progression, inhibit apoptosis, protect mitochondria, and modulate DNA damage response [108]. Some of these functions are not linked to the enzymatic activities of TERT, but to intracellular interactions of TERT, specifically activation of Wnt/β-catenin [108] and of NF-κB signaling pathways [109,110]. Mechanism(s) behind these effects of TERT are yet unknown and may be linked to interaction of TERT with specific cellular partners such as Myc involved in numerous cell-signaling cascades. Direct protein–protein interaction stabilizes Myc and promotes its transcription activity, including the protooncogenic ones. In the absence of TERT, Myc is degraded by the proteasome (as in normal cells). In malignant cells expressing TERT, interaction of TERT with Myc through its MBI domain stabilizes both proteins and promotes protooncogenic activity of Myc supporting proliferation of malignant cells, in part by direct activation of TERT transcription [111]. Inability of cells to degrade TERT together with TERT-stimulated cell proliferation are the prerequisites of TERT-induced cell immortalization. These properties (interaction with Myc with its stabilization and interference with cell signaling) are not abrogated by TERT inactivation. Some interactions (as interaction with Myc) involve yet unidentified domains/motives of TERT. Their “neutralization” would require a dramatic protein modification. Use of relatively short RT domain of TERT could be a solution. Besides, our data indicate that RT domain of TERT may act not only as an immunogen but also as a source of RNA and RNA/DNA triggering production of type I IFNs, i.e., provide in-built adjuvant(s) triggering the desired type of innate immune response against cancer cells.

## 5. Conclusions

In conclusion, topical, antigen/epitope specific retargeting of antitumor immune response may pave a way to new cancer vaccines. With this study, we have shown that it can be achieved by using single TAA domains, circumventing the complexity and variability of the full-length TAAs. This will also help to reduce an unwanted/excessive autoimmune response which could damage the host and support tumor growth. Such “module” vaccines can be introduced both systemically and intratumorally (by intratumoral DNA immunization or viral transduction). Such “tagged” tumors would then be made visible to the immune system trained by cancer vaccine. Intratumoral administration of rtTERT could also trigger production of type I IFNs, change tumor microenvironment, aiding to development of efficient adaptive antitumor response. We plan to test this approach in application to TERT in murine cancer models.

## Figures and Tables

**Figure 1 vaccines-08-00318-f001:**
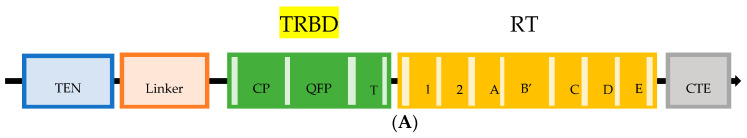
Domain structure (**A**), evolutionary tree of rat, murine and human telomerase reverse transcriptase (TERT) proteins (UniProtKB, Q673L6, O70372, and O14746, respectively) (**B**); and regions of rat TERT containing clusters of T cell epitopes recognized by immune system of mice and humans: aa 351–400 in the TERT oligomerization domain, containing part of the ciliate-specific motif CP (**C**); aa 520–620 (**D**); aa 790–870 containing active center of telomerase reverse transcriptase LVVV required for nucleotide incorporation and primer extension (**E**); aa 891–940 containing motif required for oligomerization and sequence WCGLL responsible for primer grip (**F**); aa 971–1010 containing part of the C-Terminal Extension domain CTE (**G**). The domain structure of TERT includes: TEN—telomerase essential N-terminal domain, CTE—C-terminal extension; TRBD—telomerase RNA-binding domain; and RT—reverse transcriptase domain. Domain information is given according to UNIPROT (www.uniprot.org/uniprot/O14746) and review by Rubtsova M.P. et al. [52]. The evolutionary history was inferred using the neighbor-joining method [53]. The evolutionary distances were computed using the p-distance method [54] and were presented in the units reflecting the number of amino acid differences per site. Final dataset had a total of 1109 positions. Evolutionary analyses were conducted in Molecular Evolutionary Genetics Analysis software (MEGA7) [55]. To visualize regions of rat TERT containing clusters of T cell epitopes recognized by immune system of mice and humans, epitope-rich fragments of rat TERT were aligned to the respective fragments of human TERT isotype 1 (UniProtKB # O14746.1) and mouse TERT (UniProtKB # O70372.1). Peptides representing known epitopes localized in these regions are abbreviated as “TERT” followed by the position of the first amino acid residue of the peptide according to their enumeration in rat TERT (UniProtKB #Q673L6.1) and reference to respective publications. Peptides TERT1 to TERT9 chosen for the analysis of immune response induced by TERT DNA based on the epitope analysis are outlined in bold letters in the alignment; their sequences within rat TERT (UniProtKB #Q673L6.1) are underlined or given in rectangular.

**Figure 2 vaccines-08-00318-f002:**
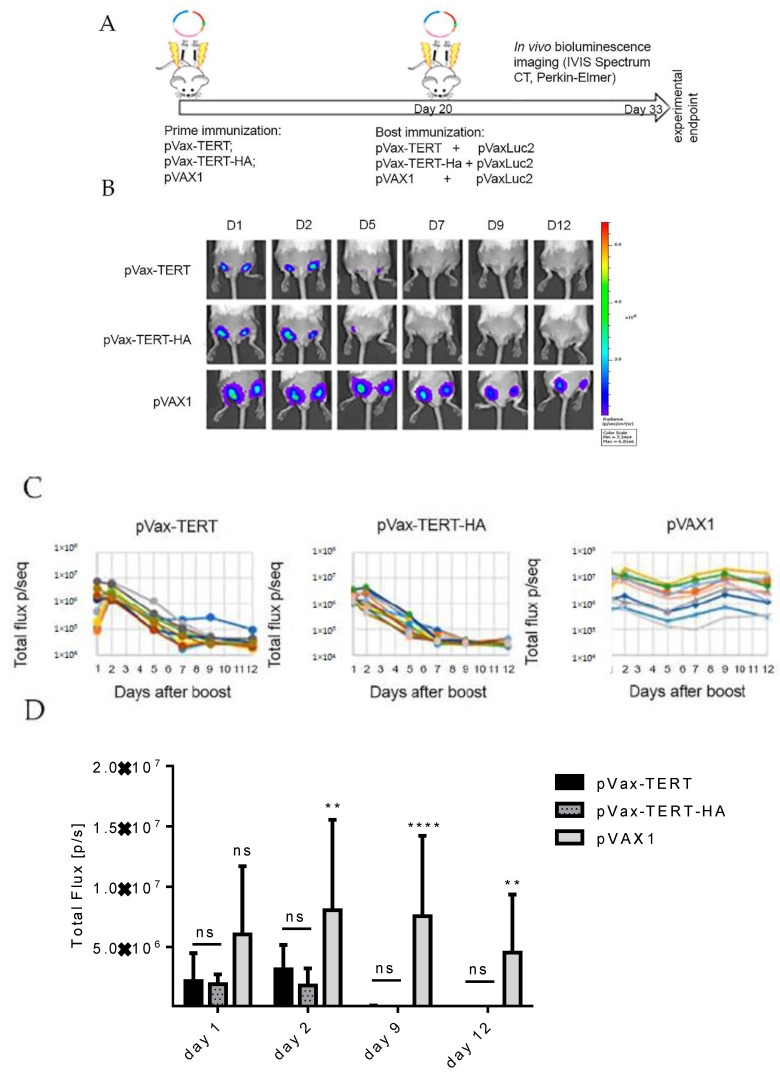
Immunization of BALB/c mice with TERT DNA, TERT variant with a C-terminal hemagglutinin tag (TERT-HA DNA), and empty vector followed by booster immunization with given plasmids mixed with DNA encoding firefly luciferase (Luc DNA), with follow up of luciferase expression by in vivo imaging. Scheme of the immunization (**A**); results of in vivo bioluminescence imaging of booster sites at days 1–12 post injection, example of 3 mice—one from TERT, one from TERT-HA, and one from empty vector group (**B**); dynamics of bioluminescence signal change in TERT, TERT-HA DNA, and empty vector-immunized mice on days 1–12 post booster injection; each line of different colors corresponds to one site of injection (two per mouse) (**C**); relative average level of bioluminescence signal for each group on days 1 to 12 post booster injection (**D**). Bioluminescence signal is represented by the total flux from site of immunization, mean ± SD. Analyzed by ordinary two-way ANOVA with Dunnett’s multiple comparison test, ** −*p* < 0.01; **** −*p* < 0.0001; ns—not significant.

**Figure 3 vaccines-08-00318-f003:**
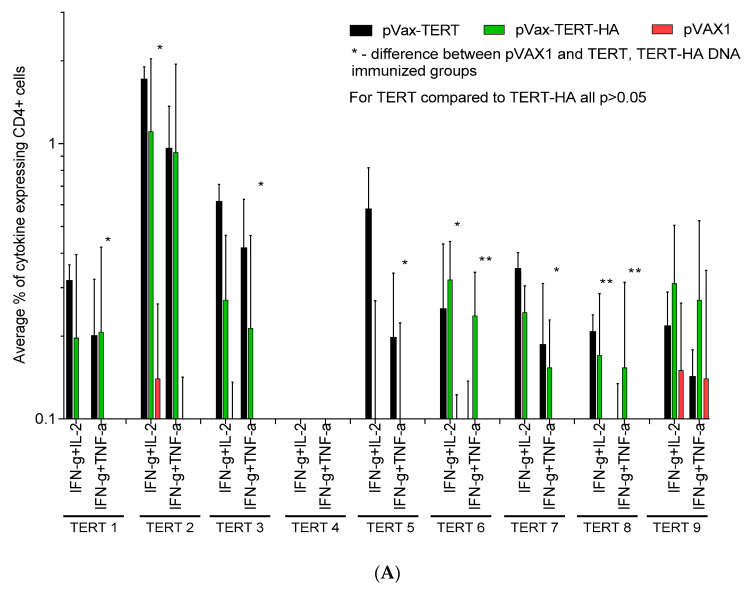
Immune recognition of TERT-derived peptides (Table 1) by CD4+ and CD8+ T cells of mice DNA immunized with TERT or TERT-HA compared to vector-immunized mice analyzed by multiparametric flow cytometry. Percent of CD4+ (**A**) and CD8+ T cells (**B**) cells reacting to stimulation with TERT peptides by double cytokine production; percent of CD4+ and CD8+ T cells reacting with triple cytokine production (**C**); and cytokine production by CD4+ and CD8+ T lymphocytes stimulated with phorbol 12-myristate 13-acetate (PMA) (**D**). Values represent mean of all mice in each group ± SD. Difference between TERT, TERT-HA DNA-immunized, and control vector-immunized mice was analyzed by Mann–Whitney test. Difference between TERT, TERT-HA DNA-immunized, and control vector: * −*p* < 0.05; ** −*p* < 0.01. No difference between TERT and TERT-HA DNA-immunized mice was found, all *p* > 0.05.

**Figure 4 vaccines-08-00318-f004:**
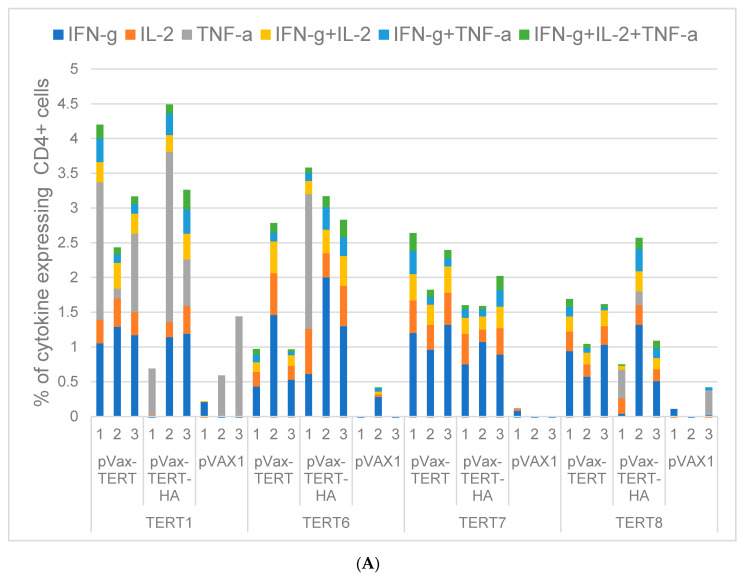
Immune recognition of peptides TERT 1, 6, 7, and 8 (Table 1) (**A**,**B**) and mitogen PMA (**C**,**D**) by splenocytes of individual TERT DNA, TERT-HA DNA, and vector-immunized mice, represented as a pile-up of the average percent of CD4+ (**A**,**C**) and CD8+ T cells (**B**,**D**) responding to in vitro antigen stimulation by production of only one, or only two, or only three cytokines (IFN-γ, IL-2, TNF-γ or IFN-γ/IL-2, IFN-γ/TNF-γ or IFN-γ/IL-2/TNF-γ, respectively, i.e., cell populations are nonoverlapping).

**Figure 5 vaccines-08-00318-f005:**
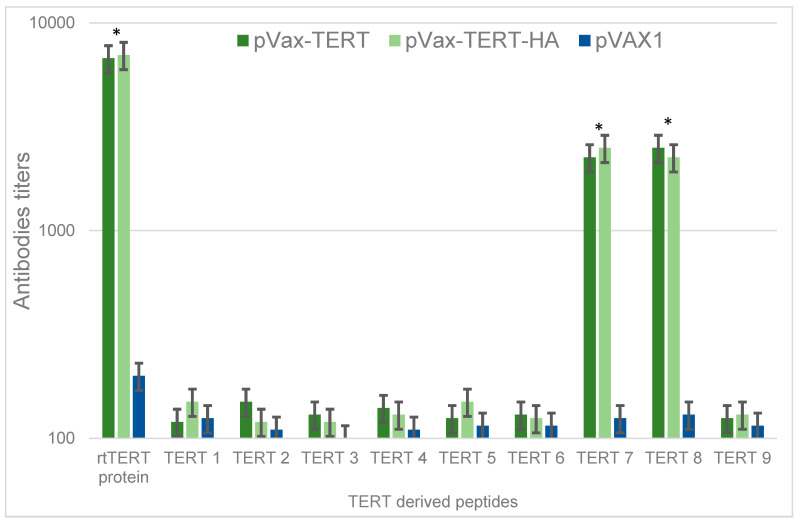
Recognition of the recombinant rtTERT and TERT-derived peptides by pooled sera of mice DNA-immunized with TERT and TERT-HA as compared to vector-immunized mice. Values represent average end point antibody titers of pooled sera in two independent ELISA runs performed in duplicate, with STDEV. *, significant difference between titers in TERT/TERT-HA DNA-immunized mice and control mice, *p* < 0.05.

**Figure 6 vaccines-08-00318-f006:**
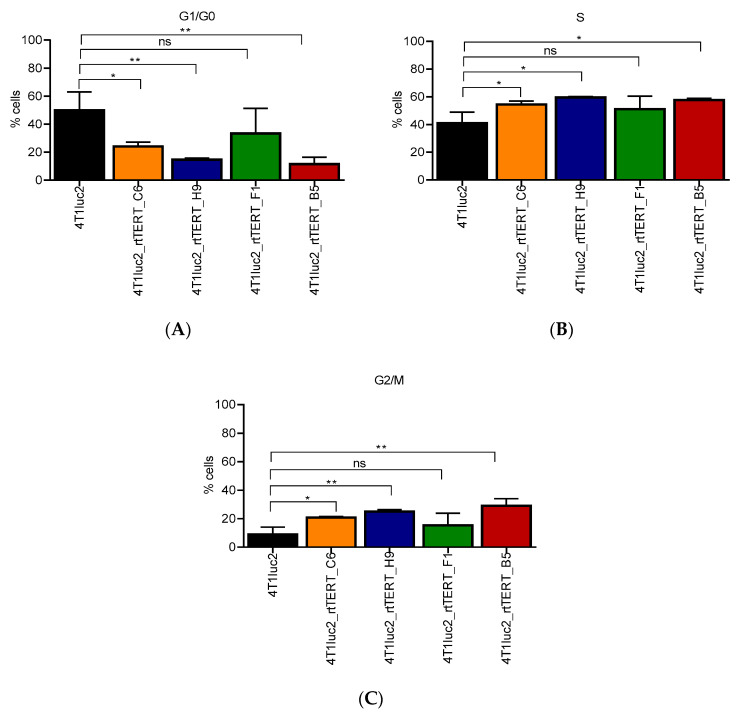
Cell cycle distribution for daughter clones of 4T1luc2 expressing *rtTERT*: G1/G0 phase (**A**), S phase (**B**), and G2/M phase (**C**). Distribution of cells in G1/G0, S, and G2/M areas was assessed using Watson pragmatic algorithm [46] in the NovoExpress software. In control groups, peak G1 was determined manually according to the manufacturer’s recommendations and peak G2 was set at ×1.75 of G1 peak. Further analysis was performed automatically with preset G1 and G2 peaks and CVs at all samples. Data were analyzed using two-tailed *t*-test, * *p* < 0.05; ** *p* < 0.01; ns—not significant.

**Figure 7 vaccines-08-00318-f007:**
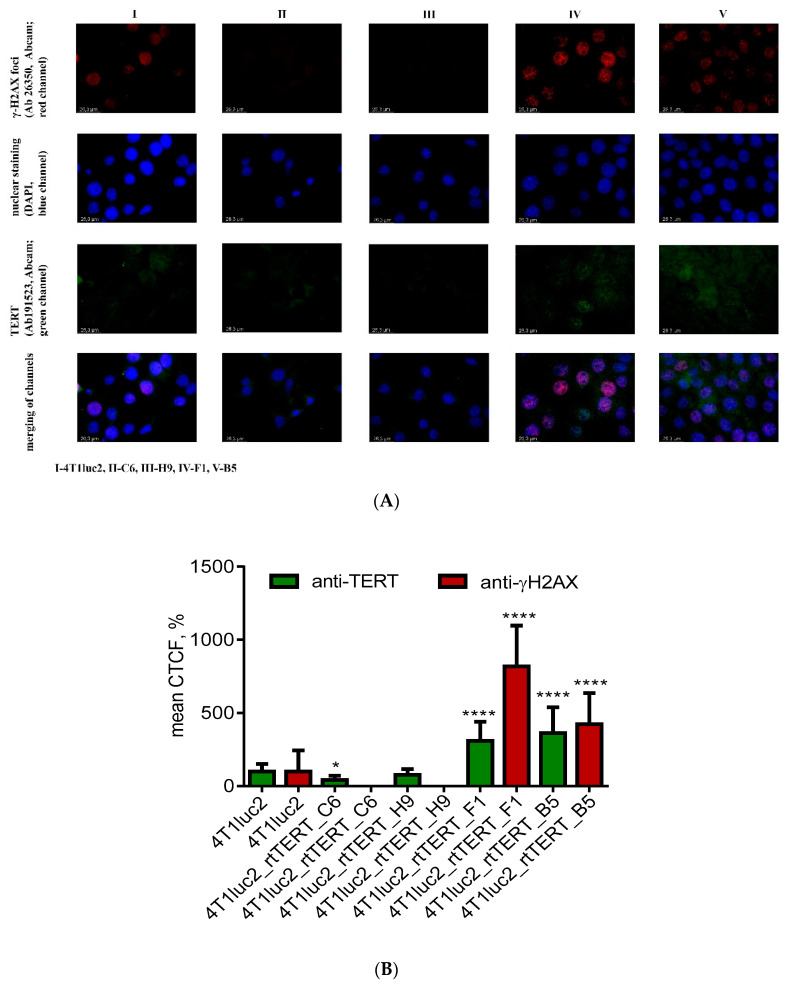
Fluorescent microscopy of 4T1luc2 cells (Panel **I**) and daughter clones expressing rtTERT: 4T1luc2_rtTERT_C6 (**II**), 4T1luc2_rtTERT_H9 (**III**), 4T1luc2_rtTERT_F1 (**IV**), and 4T1luc2_rtTERT_B5 (**V**). Staining of γ-H2AX foci (Ab 26350, Abcam; red channel); nuclear staining (DAPI, blue channel); TERT (Ab191523, Abcam; green channel); merging of channels (**A**); corrected total cell fluorescence (CTCF) for anti-γ-H2AX (red signal) and anti-TERT (green signal) relative to that exhibited by 4T1luc2, in percentage (**B**). For each cell line, at least five microscopic fields were assessed, and the average CTCF generated by specific staining were counted. CTCF was calculated for all cells as described in Materials and Methods. Results were analyzed using Kruskal–Wallis test with Dunn’s multiple comparison test. * *p* < 0.05; **** *p* < 0.0001.

**Figure 8 vaccines-08-00318-f008:**
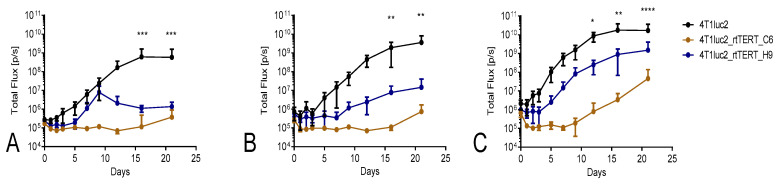
Generation of tumors by 4T1luc2 clones expressing rtTERT. Tumor growth rate was assessed using total fluorescence signal from the site of injection of 2500 (**A**), 5000 (**B**), and 10,000 (**C**) cells. Tumor volume was evaluated by total fluorescence signal from the site of cell injection by day 16 (**D**) or by calipering at day 21 (**E**). Histochemical characterization of the solid tumors formed by the parental 4T1luc2 cells (**F**) and their derivatives expressing rtTERT 4T1luc2_rtTERT_C6 (**G**); 4T1luc2_rtTERT_H9 (**H**) after ectopic implantation into BALB/c mice (H&E staining, magnification 200×). Results of tumor growth (**A**–**C**) were analyzed using RM two-way ANOVA with Dunnett’s multiple comparison test. Data on tumor volumes (**D**,**E**) were analyzed using Kruskal–Wallis with Dunn’s multiple comparison test. * *p* < 0.05; ** *p* < 0.01; *** *p* < 0.001; **** *p* < 0.0001; ns—not significant.

**Figure 9 vaccines-08-00318-f009:**
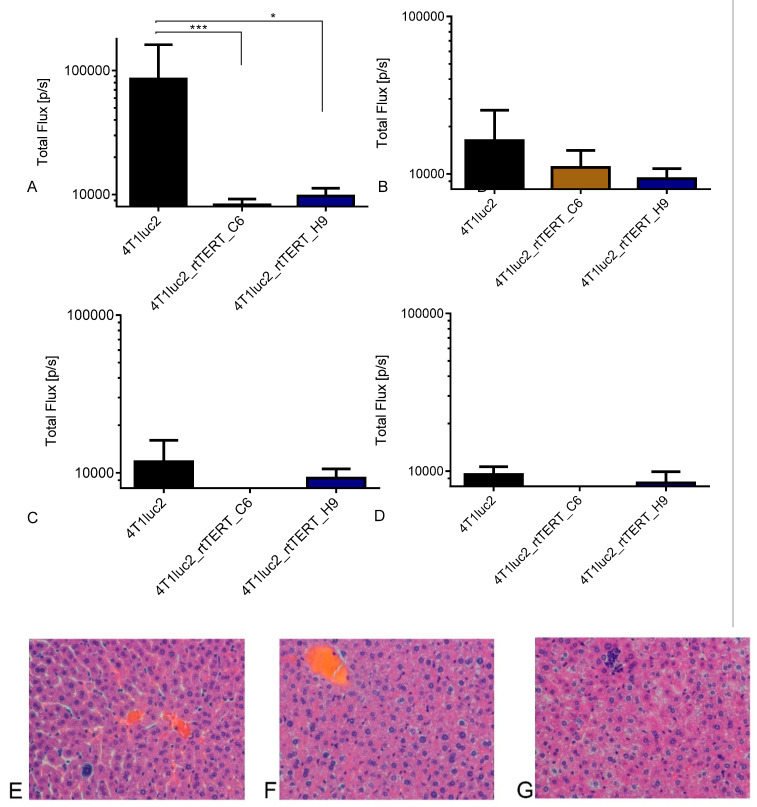
Assessment of the level of infiltration of Luc-expressing tumor cells into the organs of BALB/c mice ectopically implanted with 4T1luc2 and derivative clones 4T1luc2_rtTERT_C6 and 4T1luc2_rtTERT_H9 expressing rat rtTERT. Infiltration of organs by Luc-expressing tumor cells assessed by ex vivo BLI of lungs (**A**), liver (**B**), spleen (**C**), and kidneys (**D**). Values represent the mean total flux (p/s) ± SD (*n* = 6). Histochemical characterization of liver metastases formed by the parental 4T1luc2 cells (**E**) and their derivatives 4T1luc2_rtTERT_C6 (**F**) and 4T1luc2_rtTERT_H9 (**G**); H&E staining, magnification 400×. Comparison of the average number of liver metastases (**H**), each figure (red triangle, blue rectangular and black circle) represents single mouse), average size of liver metastases, in µm^2^ (**I**), each figure (red triangle, blue rectangular and black circle) represents single metastase), and average nn of leukocytes infiltrating the liver (**J**), each figure (red triangle, blue rectangular and black circle) represents single mouse). Number and size of metastases and number of infiltrating leukocytes were calculated in 15 high power (400×) microscope fields of hematoxylin–eosin-stained slides by computer-assisted morphometry. Data were analyzed by Kruskal–Wallis test followed by Mann–Whitney test. * *p* < 0.05; ** *p* < 0.01; *** *p* < 0.001; ns—not significant.

**Figure 10 vaccines-08-00318-f010:**
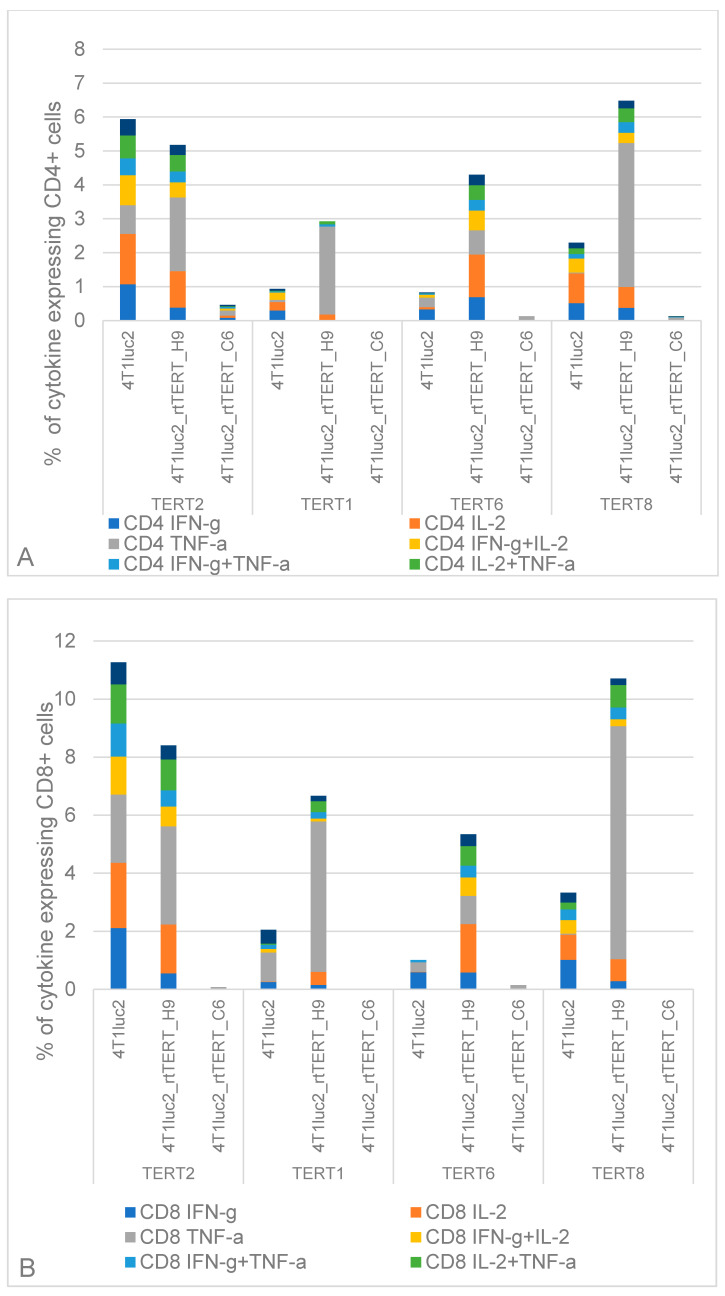
Immune recognition of TERT derived peptides by splenocytes of mice implanted with 4T1luc2, 4T1luc2_rtTERT_H9, or 4T1luc2_rtTERT_C6. Immune recognition is represented as percent of CD4+ (**A**) and CD8+ T cells (**B**) responding to stimulation with TERT 1, 2, 6, and 8 (Table 1) by production of IFN-γ, IL-2, TNF-α, IFN-γ/IL-2, IFN-γ/TNF-α, and IFN-γ/IL-2/TNF-α registered by multiparametric flow cytometry. Graphs show a pile up of the average percent of cytokine secreting CD4+ and CD8+ T cells of four mice per group, assessed in two independent runs taking two mice from each group.

**Table 1 vaccines-08-00318-t001:** Synthetic peptides used in assays of cellular and antibody responses induced by DNA immunization with rat telomerase reverse transcriptase (TERT).

Name	Amino Acid (aa) Sequence	Position in Rat TERT, 1st and Last aa Residue	Identity to Mouse TERT, Common aa/Total aa (%)	Identity to Human TERT, Unique aa/Total aa (%)	Class I Score for H2 Db Mice (IEDB)	Class I Score for H2 Dd Mice (IEDB)	The Highest Class II Scores (IEDB)	T Cell Epitopes Localized in the Region
TERT 1	PPSLTGARRLVEIIFLGSRPRTSGPFC	356–382	24/27 (89)	19/27 (70)	0.47	0.58	1.28–1.33	[30,31]
TERT 2	ILAMFLFWLMDTYVVQLLRSFFYITETT	530–558	26/28 (93)	22/28 (79)	0.3727	0.349	6.48–9.01	[30,32,33,34,35,36]
TERT 3	QKNRLFFYRKSVWSKLQSIGIRQQL	559–584	24/25 (96)	24/25 (96)	−0.185	−0.268	2.65–7.13	[30,34,35,36,37,38]
TERT 4	DTWLAMPICRLRFIPK	600–616	16/16 (100)	7/16 (44)	0.31	0.19	9.44	[32,39]
TERT 5	SLLHFFLRFVRHSVVKIDGRFYVQ	791–815	16/24 (67)	12/24 (50)	0.379	0.4973	9.4	[30,36]
TERT 6	QQDGLLLRFVDDFLLVTPHL	845–865	19/20 (95)	17/20 (85)	0.32	0.438	>10	[33,34,35]
TERT 7	KTVVNFPVETGALGGAAPHQLPAHCLFPW	888–917	26/29 (90)	22/29 (76)	0.2196	0.5080	7.75–10.73	[30,31,37]
TERT 8	LGGAAPHQLPAHCLFPWCGLLLDTRTLE	901–929	26/28 (93)	23/28 (82)	0.355	0.333	11.25	[30,31,33,37]
TERT 9	FLDLQVNSLQTVCINIYKIFLLQAYRFHACVI	973–1001	32/32 (100)	29/32 (90)	0.176	0.1579	7.81–10.77	[30,31,33,34,35,36,38]

**Table 2 vaccines-08-00318-t002:** Characterization of daughter clones of 4T1luc2 expressing rtTERT by the copy number of *rtTERT* inserts and by the expression of *rtTERT* mRNA. Copy number of *rtTERT* was assessed by digital droplet PCR (ddPCR) in relation to *Mstn* and *Actb*. Levels of *rtTERT* mRNA were normalized to *HPRT1* and calculated as fold change compared to 4T1luc2_rtTERT_B5 clone characterized by the lowest level of *rtTERT* expression. Data represent an average of at least two independent assays. Nn: number, N/A: not applicable.

Clones of 4T1luc2 Cells with rtTERT Inserts, Full Name	Abbreviated Name	Nn of Genomic Copies of *rtTERT* by *Actb*	Nn of Genomic Copies of *rtTERT* by *Mstn*	*rtTERT* mRNA Expression Relative to *HPRT1*
4T1luc2_rtTERT_B5	B5	0.57 ± 0.08	0.45 ± 0.02	1.00 ± 0.00
4T1luc2_rtTERT_C6	C6	1.33 ± 0.14	1.05 ± 0.09	6.56 ± 1.63
4T1luc2_rtTERT_H9 *	H9	2.96 ± 0.24	1.66 ± 0.19	15.28 ± 1.72
4T1luc2_rtTERT_F1	F1	0.70 ± 0.06	0.70 ± 0.06	5.95 ± 0.70
4T1luc2	4T1luc2	N/A	N/A	N/A

* No difference between nn of genomic copies of rtTERT determined using Actb and Mstn reference genes for all clones except 4T1luc2_rtTERT_H9 *.

## Data Availability

All data used to support the findings of this study are included within the article and the supplementary information file(s). Data not shown in the article used to support the findings of this study are available from the corresponding author upon request.

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
