# Peer review of "Expression of the Reverse Transcriptase Domain of Telomerase Reverse Transcriptase Induces Lytic Cellular Response in DNA-Immunized Mice and Limits Tumorigenic and Metastatic Potential of Murine Adenocarcinoma 4T1 Cells"

_vaccines, 2020, doi:10.3390/vaccines8020318_

Round 1

Reviewer 1 Report

Dear Authors, 

The manuscript titled "Expression of the reverse transcriptase domain of telomerase reverse transcriptase induces lytic cellular response in DNA-immunized mice and limits tumorigenic and metastatic potential of murine adenocarcinoma 4T1 cells" is well written and logically constructed.

I have just few suggestion to improve the quality of the paper. 

1-please read again, carefully the manuscript in order to avoid any typos (such as no spaces between two words; write the same word in the same way (E. coli and not E coli) and other.

2- the histogram should be better showed, and the font should be the same in legends, X and Y axis titles, and the different histogram should have the same format. 

3- since angiogenesis is one of the tumor hallmark, have the Authors data on this field? 

4- have the Authors data concerning the tumor progression after 1 month or more time? 

Author Response

Authors wish to thank Reviewer 1 for positive evaluation of the study. Below, please, find our response to the suggestions aimed to improve the quality of the paper:

1-please read again, carefully the manuscript in order to avoid any typos (such as no spaces between two words; write the same word in the same way (E. coli and not E coli) and other.

Manuscript was revised to avoid spelling mistakes and typos, towards homogenization of the terms and abbreviations used.

2- the histogram should be better showed, and the font should be the same in legends, X and Y axis titles, and the different histogram should have the same format.

We have revised the figures to adhere to the same font in legends, X and Y axis, although we could not keep the same font size everywhere, due to space restrictions. The main font size used throughout the text and in all figures was 12, however, in some of the figures, X axis labels were long and could not be placed in full at this font size, in this case, we reduced the font size to 9.  We adhered to the same format in histograms that belong to different panels within one and the same figure.

3- since angiogenesis is one of the tumor hallmark, have the Authors data on this field? 

Within this study, we limited ourselves to assessing tumor size and metastasis formation on the basic histochemical level (H&E stainings) and did not study angiogenesis. We thank the Reviewer for the comment and will include such survey into our coming study.

4- have the Authors data concerning the tumor progression after 1 month or more time?

Principle that we adhered to in the described study was to terminate the experiment when the first tumors formed (normally, ones in the control group of mice implanted with 4T1luc2 cells) reach 1 cm(3), in lines with the ethical permit. After that, experiment was terminated and all mice were assessed for tumors and metastasis in the same way. 1 cm(3) tumors in control mice are normally formed by day 21-22 of the experiment (before 1 month). With this experimental layout we could not follow mice for longer times.

Formally, one can change the end-point, and set it to reaching 1 cm(3) for each mice in the experiment. This approach has several draw backs. Firstly, it requires individualized approach as all mice have then to be screened regularly with short intervals, not to miss time when the tumor reaches 1 cm(3). Secondly, some mice in the cage will have to be sacrificed, while others will continue, which may result in too few, up to single mice, alive in each cage. Homing of single/few mice in one cage is not recommended as causing stress and discomfort for the animals.

In principle, this comment of the Reviewer raises a very important question - if few cancer cells may survive at the implantation site and give rise to the tumor at the later stages. We are currently planning such experiment with different controls which would allow to make longer follow ups.

Reviewer 2 Report

The authors have investigated the immune response elicited by rtTERT in DNA immunized mice and in 4T1adenocarcinoma cells. The work is pretty extensive and well designed. The text needs to be rewritten to make it more clear for readers to follow. 

  1. The english language needs to be checked for grammatical errors. This will help readers better follow the text
  2. The introduction needs to include more details on reverse transcriptase and how it controls cell proliferation. The authors should also include how this process is hijacked by tumor cells
  3. Figure 1A and 1B are not explained in the results section of the text and this needs to be included
  4. FACS plots showing enrichment of CD8+ and CD4+ subpopulations needs to be included
  5. In figure 7 the authors should label each channel (DAPI/DNA etc) next to the image panel. 

Author Response

Authors wish to thank the Reviewer for the careful assessment of the results of the study and positive evaluation of the results. Please, find below our response to the critical notes.

1- The english language needs to be checked for grammatical errors. This will help readers better follow the text

We have revised the manuscript to eliminate grammatical errors and typing mistakes.

2- The introduction needs to include more details on reverse transcriptase and how it controls cell proliferation. The authors should also include how this process is hijacked by tumor cells

We have included into introduction a paragraph describing the enzymatic activity of TERT, how it controls cell proliferation and how this process is hijacked by tumor cells. Additional details on TERT activity is given in conjunction with the discussion of the results of the study.  

3- Figure 1A and 1B are not explained in the results section of the text and this needs to be included.

Figures 1A and B are referred to in the Results, we substantiated the reference with text referring to figure content. Figures 1A shows domain structure of TERT to visualize the localization and size of the reverse transcriptase domain on which we further concentrate in relation to other TERT domains and motives. Figures 1B shows the degree of homology of TERT proteins derived from various species, to show that there is a difference between TERT of rat and TERT of mice and men, allowing us to expect overcoming of tolerance in preclinical and clinical vaccine trials, respectively.

4- FACS plots showing enrichment of CD8+ and CD4+ subpopulations needs to be included.

We have Figure in the supplement illustrating gating procedure. It is made to show gating for CD4+ and for CD8+ (enrichment; Supplementary Fig.5). It was referred to in the Material and Methods section, we referred to this figure also in the Results.

5- In figure 7 the authors should label each channel (DAPI/DNA etc) next to the image panel.

We have modified Figure 7 and instead of coding panels, labelled them with the channel directly in the body of the figure.